



# South American regional smoke plume in recent years: main sources and impact on solar radiation focusing on the Pantanal 2020 biomass burning season

Nilton Évora do Rosário[1], Elisa Thomé Sena[2], and Marcia Akemi Yamasoe[3]

[1]Departamento de Ciências Ambientais, Universidade Federal de São Paulo, Diadema, Brazil
[2]Departamento Multidisciplinar, Universidade Federal de São Paulo, Osasco, Brazil
[3]Departamento de Ciências Atmosféricas, Instituto de Astronomia, Geofísica e Ciências Atmosféricas, Universidade de São Paulo, São Paulo, Brazil

**Correspondence:** Nilton M. Évora do Rosário (nrosario@unifesp.br)

**Abstract.** The 2020 biomass burning season in Brazil was marked by an unprecedented amount of fire counts across the Pantanal biome, which led to high levels of air pollution within the biome and downwind areas. Large amount of fire counts was also detected in the Amazon Forest during 2020 compared with the recent years. However, the contribution of Pantanal fire emissions to the regional smoke plume was speculated to rival the contribution of fire emissions from Amazon. Aiming to

contextualise the 2020 biomass burning season focusing on the unprecedented role of Pantanal, the present study's main goal is to analyse the recent biomass burning seasons in Brazil looking at the fire counts, the regional smoke plume and its impact on surface solar radiation (SSR). The focus is on the biomes most affected by the recent biomass burning events, Amazon forest, Cerrado and especially Pantanal. To characterise the regional smoke plume we analysed aerosol optical depth, single scattering albedo and its impact on the solar radiation reaching the surface. The influence of interannual variability of the wind

at 850 hPa on the transport of the regional smoke plumes was also explored. In 2020, the regional smoke plume covered an area well above 6 million km2, the largest area in the last six years, but equivalent to the observed in a more remote past, as in 2007 and 2010. However, from the point of view of Pantanal, 2020 was an unprecedented year, not due to the amount of smoke over the biome, but regarding the biome contribution to the regional smoke plume. The number of fire counts was 3.4 times higher than the mean value considering the period from 2003 to 2020. The entire biome was continuously covered

by a thick smoke layer from September to October, which resulted in a monthly mean deficit of surface solar radiation up to 300 Wm-2. Additionally, the 2020 regional smoke plume presented higher optical absorption when compared with the recent years plumes, which could be related to the Pantanal larger fire emission. However, current knowledge on optical and radiative properties of smoke aerosols from Pantanal is limited compared to the one resultant from Amazon and Cerrado fire emissions, which prevent a definitive conclusion.



# 1 Introduction

In South America, the regional biomass burning smoke plume is the most important signature of anthropogenic activities from the point of view of injection of pollutants into the atmosphere at continental scale during the dry season (Prins and Menzel, 1992, Artaxo et al., 1998, Freitas et al., 2004, Longo et al., 2007). Its geographical dimension and pollutants loading, along with its climate and air quality effects, have been comprehensively studied during the last decades (Freitas et al., 2004, Ignotti


et al., 2010, Artaxo et al., 2013, Rosario et al., 2013, Chen et al., 2013, Sena et al., 2015, Moreira et al. 2017, Thornhill et al., 2018). Its perturbation on the regional climate span from lowering the solar energy availability (Procópio et al., 2001, Schafer et al., 2002, Yamasoe et al., 2006, Rosario et al., 2013, Moreira et al. 2017) for surface processes (biological, physical and chemical) to cloud microphysics and atmospheric thermodynamics and chemistry. In the context of air quality, especially near biomass burning areas, its impact on air pollution levels surpasses the most polluted urban areas of the continent (Ignotti et


al., 2010, Sena et al., 2013, Rosario et al., 2013). Systematically, along the years, the southern portion of the Amazon forest and the eastern part of the Cerrado ecosystem have been the major sources of pollutants to the Regional Smoke Plume (RSP). Consequently, in general, these are the most affected areas by the smoke pollution (Artaxo et al., 2013, Pereira et al., 2016), in particular, when compared with the remaining Brazilian biomes, Pantanal, Caatinga, Mata Atlantica and Pampas. From the mid 2000s to the earliest years of the 2010 decade, following the trends observed in deforestation and fire counts in Amazonia


and Cerrado, the RSP loading and dimension were significantly reduced (Reddington et al. 2015). However, in recent years, aerosol loading levels in the RSP reached values observed at the peak of the deforestation and fire spot counts from a relatively remote past. The 2020 biomass burning in Brazil, amid the COVID-19 pandemic, has been claimed to be one of the worst in recent years, indeed an unparalleled biomass burning season, at least from the point of view of the Pantanal biome. How does the 2020 biomass burning season in Brazil compare with the previous that occurred in recent years? Unlike what has


been seen historically, the fire counts in Pantanal is claimed to have played a major role in 2020 biomass burning, according to the environmental agencies, institutes and main media headlines across the country and worldwide (WWF, 2020, FSP, 2020, BBC, 2020, NYT, 2020, Le Monde, 2020, FSP, 2021). To what extent the 2020 biomass burning season and the produced RSP surpassed the previous recent years? Was there an unprecedented signature of Pantanal biome biomass burning season in the 2020 RSP? These are some of the questions that the present analysis addressed in order to contextualise the 2020 biomass


burning season and explore the role of Pantanal biome regarding the RSP dimension, loading and impact on surface solar radiation. From here on, the article is organised as follow: Section 2 presents a brief description of the study region, a summary of the data used and the methods adopted; Section 3 presents the results divided in 3 sub-sections: Sub-section 3.1 analyzes the intraseasonal and interannual variability of fire counts, aerosol optical depth (AOD) at 550 nm, downward solar radiation at the surface for the three main biomes Amazon tropical rainforest, Cerrado ecosystem, and Pantanal, for the last six years.


Sub-section 3.2, focuses on September, the month when biomass burning typically peaks in Brazil, providing an interannual analysis of the spatial distribution of AOD, wind field at 850 hPa and downward solar radiation at the surface across the selected biomes under cloudless conditions. In the last sub-section, Sub-section 3.3, the analysis is focused on a long term interannual





and intraseasonal variability of AOD, fire counts and surface solar radiation over Pantanal. In Section 4, the main conclusions
are summarized.

## 2    Study Region, Data and Methods

This study focused on three major Brazilian biomes: Amazon tropical forest (or Amazonia), Cerrado ecosystem and Pantanal
wetlands (Figure 1). According to Pivello (2011), the Amazon tropical forest is a fire-sensitive ecosystem and cannot tolerate
fire, with consequent mortality of trees after repeated fires. Although Pantanal and Cerrado are considered fire-dependent
biomes and, therefore, are more adapted to fire occurrence, human-driven fire frequency increased significantly in Brazil, which
can lead to land degradation and biodiversity loss (Pivello, 2011). To map the smoke plume spatial dimension and loading,
monthly mean aerosol optical depth taken from MODIS (Moderate Resolution Imaging Spectroradiometer) atmosphere Level-
3 products from Aqua satellite (MYD08), at 550 nm, and from AERONET (Aerosol Robotic Network), at 500 nm, ground-
based solar photometers were analyzed. The regional plume border definition was somehow arbitrary, since the dispersion of
the smoke is a continuous process and it is impossible to track the effective ending of its influence. To provide a perspective
on the plume spatial dimension and loading at its core, isolines of AOD at 550 nm of 0.3 were used as reference to evaluate
the plume dimension. The value of 0.3 was found convenient since it is above typical monthly average values of AOD at
550 nm observed over major urban areas of South America, which prevent a misidentification related to these urban plumes.
Based on this AOD threshold to define the main area of the plume, the RSP loading reference was taken as the mean AOD
within the delimited area. The mean AOD was also calculated for the domain of the biomes traditionally affected by the plume,
Amazonia, Cerrado and Pantanal. AERONET aerosol optical depth at 500 nm (Holben et al., 1998), single scattering albedo
at 440 nm (Dubovik and King, 2000) and aerosol direct radiative forcing at the surface (Garcia et al., 2012), level 1.5, from
three sites located in different parts of Brazil and one in Bolivia, were included in the analysis to help identify the extension
of the RSP and differences in the aerosol optical properties from different years and locations downwind from the fire sources.
Since no level 2.0 data are available for the most recent period, we compared AERONET data from levels 1.5 and 2.0 in
Figure A.1(Appendice 1), to demonstrate the good quality of level 1.5 data as well. To better contextualise the 2020 biomass
burning season, we also analysed the interannual variability of the fire counts in each of the considered biomes and the regional
downward solar radiation at the surface. Table 1 presents the variables and their respective applications, data sources and
references. Fire counts were obtained from the Brazilian Space Agency (Instituto Nacional de Pesquisas Espaciais - INPE).
These are important data to explore the spatial dynamic of smoke emission sources and, to some extent, to analyse the extension
of illegal biomass burning, which has been the main driver of the regional smoke plume in Brazilian biomes. Fire counts are
based on Moderate Resolution Spectroradiometer (MODIS) data and two algorithms. In summary, according to Morisette et
al. (2005), the algorithms use empirically derived thresholds based on digital numbers (DNs) at channels 20 (at around 3.7
$\mu$m) and 9 (around 440 nm). In the daytime algorithm, pixels are classified as "fire" if two conditions are satisfied: DNs higher
than 3000 at channel 20 and lower than 3300 at channel 9. The nighttime algorithm requires only one condition: DNs higher
than 3000 at channel 20. To estimate the direct effect of the aerosol plume on the reduction of surface solar radiation (SSR),



instantaneous retrievals of CERES (Cloud and the Earth's Radiant Energy System) Single Scanner Footprint - Level 3 during Aqua overpasses were used. This variable is estimated by a combination of CERES SW upward irradiance at the top of the atmosphere retrievals, ancillary meteorological data, surface, aerosol, gases and cloud properties. These inputs are used in the Langley parameterized shortwave algorithm (LPSA) (Kratz et al., 2020), to estimate shortwave downward irradiance at the surface from 0.2 to 5.0 $\mu$m. The output is gridded in 1 by 1 latitude/longitude resolution. Cloud-free scenes in the presence of aerosols were selected to account only for the aerosol direct effect on radiation. The downward irradiance for the second half of August was missing due to Aqua bouncing, thus, results for August 2020 are presented but comparison with other months and years must be made with caution. Wind circulation at 850 hPa plays a determinant role on the transport of smoke over South America (Freitas et al. 2004) and, therefore, on the regional smoke structure, especially for areas downwind of the main biomass burning areas, namely southern of Amazon Forest and western portion of Cerrado ecosystem. Due to its position, Pantanal biome is subject to smoke transport from the both larger biomes, via north (from Amazon forest and Cerrado) and via east (from Cerrado). With this perspective, the interannual variability of eastward and northward components of the wind at 850 hPa considering a transect from the southern portion of Amazon forest to areas in the southern of Brazil and centred on Pantanal were analysed. The wind dataset to perform this analysis were taken from MERRA-2 (Modern-Era Retrospective analysis for Research and Applications, version 2) reanalysis (Gelaro, et al., 2017). The interannual and space variability of the fire counts, AOD and downward solar radiation at the surface were analysed, based on monthly mean values (July, August, September and October), and from the perspective of the selected biomes. Subsequently, the interannual variability of aerosol optical depth, single scattering albedo and mean radiative forcing were analysed. Those variables were also based on monthly mean, from AERONET stations located in different positions of the regional plume influence dominium. Being typically the peak of the biomass burning season in Brazil, September was selected to carry on a more comprehensive geographical and interannual analysis of the features and the impact of the regional smoke plume from the last six years, also including the analysis of the downward solar radiation at the surface. Finally, extending further in the past (2003-2020), an interannual analysis of total fire counts, mean AOD at 550 nm and downward solar radiation at the surface over Pantanal was carried out focusing on the biomass burning season months, the same months as before.

## 3   Results

### 3.1   Biomass burning season: Biomes intraseasonal and Interannual variability

This section addresses the interannual and the intraseasonal variability of fire counts and the regional smoke plume analysing its characteristics and impacts on solar radiation at the surface for the three selected biomes (Amazon, Cerrado and Pantanal) during the biomass burning season of the last six years, from 2015 to 2020. Figure 2 shows, for each biome and multiple years, the intraseasonal evolution of fire counts, aerosol loading (AOD at 550 nm), and downward solar irradiance at the surface. The typical intraseasonality of the biomass burning season is observed for most of all the six years as follows: from the transition month of July, the number of fire counts increased significantly in the following months, reaching its peak in September. After that, the number of fire counts decreased, but still remained high, in October, until the next wet season. The burned area





variability, not shown here, closely follows the variability of fire counts. During 2019, in the Amazon, the fire count peaked
in August instead of September, and that was the case for Cerrado in 2016. While 2016 and 2018 were less active years from
the point of view of fire counts in both Cerrado and the Amazon, the years of 2015, 2017, 2019 and 2020 were very active.
In Cerrado, the intraseasonal variability and the amount of fire counts were quite similar in these active years, except for 2015
which presented a more active October when compared with the typical feature. For the Amazon, the major difference between
2020 and the previous five years was the high fire activity in August, characterised by a level similar to that of September, the
climatological peak month. It should be noted that among the analyzed years, 2017 presented the highest fire count in Cerrado
and Amazonia at the peak of the BB season. However, when one looks at the Pantanal biome scenario, 2020 was indeed an
extreme BB season, with almost three and four times the fire counts typically observed in August and September, respectively.
On top of that, the previous year, 2019, was already a record breaking year, when considering the previous 4 years. The
2020 extreme fire counts scenario for Pantanal is clearly seen in aerosol mean loading over the biome, with twice the values
observed in the earlier years. The result of the observed high amount of aerosol over Pantanal can also be evaluated from the
unprecedented low amount of solar radiation at the surface in October, when compared to the previous 5 years. Mean aerosol
loadings for Amazon and Cerrado in the 2020 BB season were among the highest values observed in recent years but they
were not substantially, or even, higher than the highly polluted previous years. For instance, in September of 2017, the aerosol
loading over Amazon was higher than in 2020, and for Cerrado, the September 2015 mean AOD was similar to that of 2020.
At this point, it is worth mentioning that the difference in the total area occupied by each biome must be taken into account.
It is also worth pointing out that there is a strong gradient in fire counts and smoke distribution across both the Amazon and
Cerrado biomes, being the southern portion of the former and the westernmost portion of the latter mostly affected, following
what is called the deforestation belt (Fearnside et al., 2009, Braghiere et al., 2020). Over these specific portions of the biomes,
aerosol loadings are usually much higher. Regarding the amount of solar energy at the surface, the monthly mean value for
the Amazon, at the peak of 2020 BB season, is among the lowest values, only comparable to the value observed in 2017 BB
season, and consistent with the AOD behavior. An interesting aspect to point out is that, for the Cerrado biome, the years of
2017 and 2020 presented the highest amount of solar radiation at the surface, nevertheless fire counts were among the highest
observed values for these particular years. To summarize, this multiyear analysis across the 3 biomes definitely revealed that
Pantanal signature is the main highlight aspect in the context of 2020 regional smoke plume, since the number of fire counts
and AOD dramatically increased in this biome, compared to the previous years, especially in September and October. For
instance, in September, fire counts rose from about 2900 counts in 2019 to more than 8100 in 2020. By contrast, the amount
of solar energy reaching the ground in Pantanal was significantly reduced, from about 730 Wm-2 in October 2018 to around
600 Wm-2 in the same month of 2020. To complement this biomes-based interannual and intraseasonal analysis, available
data from the AERONET sun photometer stations distributed across the BB regional plume typical corridor were analyzed
(Figure 3). From the point of view of aerosol loading, as assessed via AOD, over Alta Floresta and Rio Branco sites, both in
the southern part of Amazon, the years of 2017, 2019 and 2020 were the most polluted. These sites are upwind of Pantanal,
therefore they typically work as source areas of smoke towards Pantanal and also Cerrado biomes. Monthly mean AOD at 500
nm over Alta Floresta and Rio Branco in 2020 was as high as 1.0, a severely polluted scenario, however it was still below the





extreme scenario seen in Alta Floresta in September 2017. Closer to Pantanal, but yet upwind of the wetland biome, the sites of Santa Cruz, in Bolivia, and Cuiabá, this later within Cerrado biome, revealed that among the analyzed seasons, the last two BB seasons, in 2019 and 2020, were the most polluted, however, with different intraseasonal patterns. In 2019 and 2020 the smoke plume loading peaked in August and October, respectively, over Cuiabá. Over Santa Cruz, the maximum in 2019 and 2020 occurred in September and October, respectively. The maximum aerosol loading over Cuiabá in October 2020 overcame the values of all months and from all sites in that year, including the peak of Alta Floresta and Rio Branco, both located in the Amazon biome, a portion strongly affected by deforestation and biomass burning. Single scattering albedo (SSA) is an intrinsic property of the aerosol particles, given by the ratio of the scattering over the extinction coefficients of the aerosol. It is a parameter commonly used to infer the aerosol capacity to absorb radiation at a particular wavelength. The lower the SSA, the more absorbing the aerosol is. As it can be observed from the AERONET SSA product over the analyzed sites (Figure 3), in 2020 for most of them, SSA was lower than in previous years, mainly in the earlier months of the BB season. Although this result is regarded as limited to be conclusive, it is worth speculating about the 2020 regional smoke plume being more absorbing than plumes from previous years. One hypothesis is that the dryness and the type of biomass burned in Pantanal during the 2020 BB season, which contributed significantly to the regional plume, produced a scenario dominated by burning in the flaming phase, a phase that has been shown in previous studies to produce a larger fraction of absorbing aerosol when compared with the smoldering phase typically seen in biomass burning of Amazon forest (Reid et al. 1998, Yamasoe et al. 2000). Being the contribution of Pantanal fire emissions to the regional smoke plume significantly higher than in the previous years, it is reasonable to expect a change in the plume mean intrinsic optical properties, mainly over those regions surrounding Pantanal and sites downwind. A second hypothesis for the lower single scattering values in 2020 is that the aerosol suffered less ageing process during the transport from Pantanal to the AERONET sites downwind of the plume, due to the shorter distance between Pantanal and these sites compared to the distance between the Amazon region and the sites. Ageing processes due to the coagulation of organic compounds, cloud processing and water vapor uptake can increase the aerosol single scattering albedo (Reid et al., 2005, Rosário et al., 2011). Nevertheless, it is worth mentioning that the AERONET data from quality level 2.0 for 2020 is not yet available, so level 1.5 was used. Therefore, future and further analysis on this matter is highly recommended in order to evaluate the posed hypotheses, although as observed in Figure A.1, there is very good agreement between levels 1.5 and 2.0 SSA retrievals. The interannual and intraseasonal variability of the aerosol monthly Mean Radiative Forcing (MRF) also suggests that 2020 was a non ordinary year when compared with the previous ones. Where available, MRF values in 2020 were slightly lower than in previous years. Nevertheless, the values indicate a stronger aerosol direct radiative forcing when compared with the previous years. In 2020, Santa Cruz and Cuiabá presented their AOD peak in October instead, which suggests that very likely, for these sites, aerosol radiative forcing peak shifted from September to October, likely as a consequence of the anomalous role of Pantanal fires in that year, and their influence in the smoke plume transport to these sites.

## 3.2   Biomass burning season peak: Spatial analysis of AOD and SSR

Climatologically, September is the peak of the biomass burning season over a large area of South America, and of Brazil in particular. Therefore, September was selected to explore the spatial distribution of the interannual variability of the regional



smoke plume dimension and loading, and its impact on the downward solar radiation at the surface. Figure 4 presents the interannual variability of the geographical distribution of the regional smoke plume for September in recent years (2015-2020).

The regional smoke plume domain (area) and mean intensity (AOD@550 nm level) are also identified, where the extension of the smoke plume is delimited by the isoline of AOD at 550 nm of 0.3 (black line). Due to the wind circulation pattern in the region, in general, the aerosol particles emitted from fires in the Amazon Forest and Cerrado ecosystem are initially transported westward reaching other South-American countries until it reaches the Andes Mountain range barrier (Freitas et al., 2004), when the wind circulation becomes predominantly meridional and southwards with a variable zonal component that

can fluctuate the transport of smoke between the southeast and the southern portions of Brazil. Going southwards, the plume can reach the Pantanal region, among other locations, but the smoke gets diluted during this transport process. Undoubtedly, the regional plumes of the years 2020 and 2017 were the most polluted in recent years. However, there are substantial differences between them: the 2020 regional plume presented a larger influence domain, well above 6 million km2, and stronger transport of smoke downwind of Amazonia and Pantanal, mainly towards the southeast of Brazil. In 2017, the plume intensity was higher

but its core (highest AOD values) was located mainly in areas upwind of Pantanal, indicating that the sources in that year were mainly distributed within the Amazon biome (as shown in the fire counts of Figure 2). The transport of smoke towards the southern portion of the continent was less dominant than in 2020. A possible explanation to the 2017 feature can be found in the zonal component of the wind at 850 hPa, which presented from east and stronger than normal from southern of the Amazon forest to the south of the Pantanal domain (Figure 8), therefore indicating a less strong meridional component, the driver of

the regional plume toward south. In contrast, during 2020 the atmospheric circulation patterns favoured a strong transport southward and then eastwards driving the regional plume over the southeast portion of Brazil and the Atlantic Ocean (Figure 4). Figures 4 and 5 also show that the lower incidence of solar radiation in cloud-free conditions occurred over the regions more severely affected by the aerosol plume, consistently with the spatial distribution of AOD, as expected. The available solar energy reaching the ground in September 2017 and 2020 was significantly reduced by the plume. Although the reduction

covers a large area in all the years, 2017 and 2020 are particularly noteworthy in terms of the severity of such events. The contrasting values of the downward solar irradiance from west to east, in these years, of more than 200 Wm-2 evidence the impact of the smoke layer. In both years, not only were the plume dimensions larger than usual, but also, most of the area was covered by a very thick smoke layer. In 2020 the mean AOD@550 nm over the Pantanal in September was abnormally high, above 1, and as shown in Figure 2 so was the fire counts. This atypical pattern made Pantanal an important source of smoke to

the 2020 regional plume, and this role is further explored in the next topic.

### 3.3   Biomass burning season: Pantanal biome historical perspective

The 2020 biomass burning season in Brazil was atypical, at least for the Pantanal biome, which certainly played an unusual role in the regional smoke plume. To contextualise the unusual scenario like the 2020 biomass burning season in Pantanal, we performed a historical analysis (2003-2020) of the total fire count, mean AOD@550 nm and the downward surface solar radia-

tion (SSR) focusing on this biome (Figure 6). There is a clear relationship between fire counts and aerosol loading throughout the years over Pantanal, which does not mean that the aerosol loading is exclusively a result of local fire emission, as it will





be shown. From 2004 to 2010, excluding the years 2006, 2008 and 2009, fire counts in Pantanal at the peak of the biomass burning season (September) surpassed the number of 3000, with the AOD level following the high level of fire counts. In 2007, during September, the highest aerosol loading was observed within the period analysed, which resulted in the largest reduc-

tion in surface solar radiation. In 2007 and the other polluted years of 2000s, and the year 2010, large AOD values were also observed upwind of Pantanal, in the southern of Amazon forest and western part of Cerrado, as can be seen in the Hovmoller diagram of AOD at 550 nm (Figure 7). This indicates the relevant contribution of smoke advection from these biomes to the high aerosol loading observed over Pantanal, especially during 2010 when local fire counts were relatively low. From 2011 to 2019, both fire counts and AOD levels in Pantanal were well below the values observed during the previous polluted years,

even when high levels of smoke were observed in the upwind regions, for instance in 2017 (Figure 7). In this year, the core of the transport toward the south was over Paraguay and characterised by a strong meridional component (Figure 4), which can also be inferred from the smoke plume effect on surface solar radiation (Figure 5). This westward displacement in the transport of smoke from regions upwind of Pantanal in 2017 can be explained by the stronger and persistent west component of the wind at the 850 hPa, that extended from the southern of Amazon to areas downwind of Pantanal (Figure 8). Regarding

the high levels of AOD in the northern part of the Amazon basin at the end of the year 2015 (Figure 7), and which did not have a significant influence on Pantanal region, it is worth mentioning that it occured during an El Nino event, when this part of the Amazon typically experiences drought scenarios and higher incidence of fire counts. Indeed, according to Marento et al. (2017), the Amazon onset of the rainy season in 2015 occurred later than normal, and the region was characterised by drought in 2016. Regarding the years characterised by less polluted scenarios in Pantanal (2011-2019), they were consistently

associated with relatively low local fire counts. This scenario changed during the 2020 biomass burning season, when the total fire counts in Pantanal surpassed all the previous analysed years. According to Marengo et al. (2021), since the beginning of the fire activity monitoring in Brazil, in 1998, the largest number of fires over Pantanal was detected in 2005 and, in 2020, it was 76% higher. The level of aerosol loading over the biome at the peak of the 2020 burning season was similar to the early polluted years, in the middle of 2000s, however, opposite to the former years, AOD in Pantanal in 2020 peaked in October,

when the largest reduction in solar radiation by aerosols was observed, and the aerosol loading upwind of Pantanal was much lower than that over the biome (Figure 7), suggesting that the biome itself was the main source of smoke in its domain, and with advection from outer regions playing a secondary role. October followed an also highly polluted September, when there was an explosion in fire counts across Pantanal. The increase in AOD over Pantanal, in mid September, occurred simultaneously with the peak of Amazonia, suggesting that at this time the smoke plume over Pantanal had also received contribution from

biomass burning emission from Amazonia (Figure 4). While the aerosol loading magnitude observed during the 2020 biomass burning over Pantanal and Brazil was not unique from historical perspective, there are relevant differences between 2020 and previous polluted years here analysed. First, the contribution of local biomass burning emission from Pantanal to the regional smoke plume rivals emissions from the Amazon forest and Cerrado ecosystems, specially for areas downwind of the wetland biome. An expected consequence would be a shift in the composition of the regional smoke plume properties (mainly optical

absorption), whose confirmation requires further studies, nevertheless dataset from AERONET stations here analysed already indicated a more absorbing regional plume than usual. There was also a noteworthy difference in the intraseasonal dynamic,





with a smoke plume stronger than normal during October, when the dry-to-wet transition period started, and whose direct radiative impact on solar radiation was the largest among the analysed years (Figure 6b).

## 4    Conclusions

The 2020 biomass burning season in Brazil attracted unprecedented attention from national and international media as well as the general society. Pantanal biome was a hotspot in this entire discussion. The wetland biome's role in the regional smoke plume has been marginal throughout the years, however, with the explosion of fire counts across Pantanal in 2020, that certainly was not the case for 2020. In this study, via comparison with the previous years, we analyzed to what extent the 2020 biomass burning season differs from previous years. We did so analyzing the fire counts, aerosol loading and its impact on surface solar

radiation over Pantanal and regionally, with a particular emphasis on the 3 main biomes: Amazonia tropical rainforest, Cerrado and Pantanal. In the last six years, from the point of view of the Amazon Forest and Cerrado ecosystem, 2020 ranked among the most fire active and polluted, along with 2017, when Amazonia and the western portion of Cerrado experienced their largest number of fire counts. An important aspect of 2017 is that the regional smoke plume transport toward the southern portion of the continent took place mostly over its neighbouring countries Bolivia and Paraguay, and a marginal amount was transported over

the southeast of Brazil. A possible explanation for that was found in an anomalous wind pattern at 850 hPa characterised by a stronger and persistent west component. For Pantanal, 2020 was a very particular year, not exactly due to the aerosol loading over the biome, since in September of 2007 the biome experienced a higher monthly mean AOD, but due to the contribution of the local fire emission to the regional smoke plume. In the 2020 biomass burning season, fire counts in Pantanal were 3.4 times higher than the mean value from 2003 to 2020. In the context of the last decade, monthly mean smoke loading over Pantanal

in 2020 was more than two times higher (AOD at 550 nm    1.0) than values observed in previous years, and comparable to values typically observed over the southern portion of Amazonia. The entire biome was continuously covered by a thick layer of smoke from west to east and from south to north for almost one month and a half, which resulted in a monthly mean deficit of solar radiation at the surface up to 300 Wm-2. The impact of this reduction of incoming solar radiation on biological and surface-atmosphere interactions processes are yet to be evaluated. Additionally, considering the plume transport towards the

highly productive central and southeast regions of Brazil, such attenuation in the amount of solar radiation at the surface limits the country's capacity of production of renewable energy based on solar radiation, mainly in its central portion, and during the dry season, when cloud cover is less frequent. Although it is too early to infer a possible pattern change in contribution of Pantanal fire emission to regional smoke plume, studies focusing on the 2020 atypical biomass burning season may shed light on potential scenarios that the region may experience in the future, when one considers the climate projections of increasing

the frequency of drought conditions. Current knowledge on the Pantanal biome fire emission dynamics and the optical and radiative properties of the resultant smoke plume is far limited compared to the biomass burning emission and optical and radiative properties of smoke plume produced in the Amazon forest and Cerrado ecosystem.



*Data availability.* All the dataset (AERONET, MODIS, CERES, MERRA-2, INPE fire count) used in this study are publicly available and can be downloaded from their respective sites provided in Table 1 of this manuscript.

*Author contributions.* N.E.R., E. S. T and M. A. Y. designed and performed the research, analyzed the data, and wrote the paper.

*Competing interests.* The authors declare that they have no conflict of interest.

*Acknowledgements.* This work was supported by FAPESP (Fundação de Amparo à Pesquisa do Estado de São Paulo) projects 2016/18438-0 and 2018/16048-6 and CNPq (Conselho Nacional de Desenvolvimento Científico e Tecnológico), process number 313005/2018-4 and Universal 421870/2018-4. We also acknowledge the AERONET, MERRA-2, Programa Queimadas-INPE, CERES and MODIS mission 295 scientists, Principal Investigators and associated NASA personnel for the production of the data used in this research effort.



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

April 3rd 2021



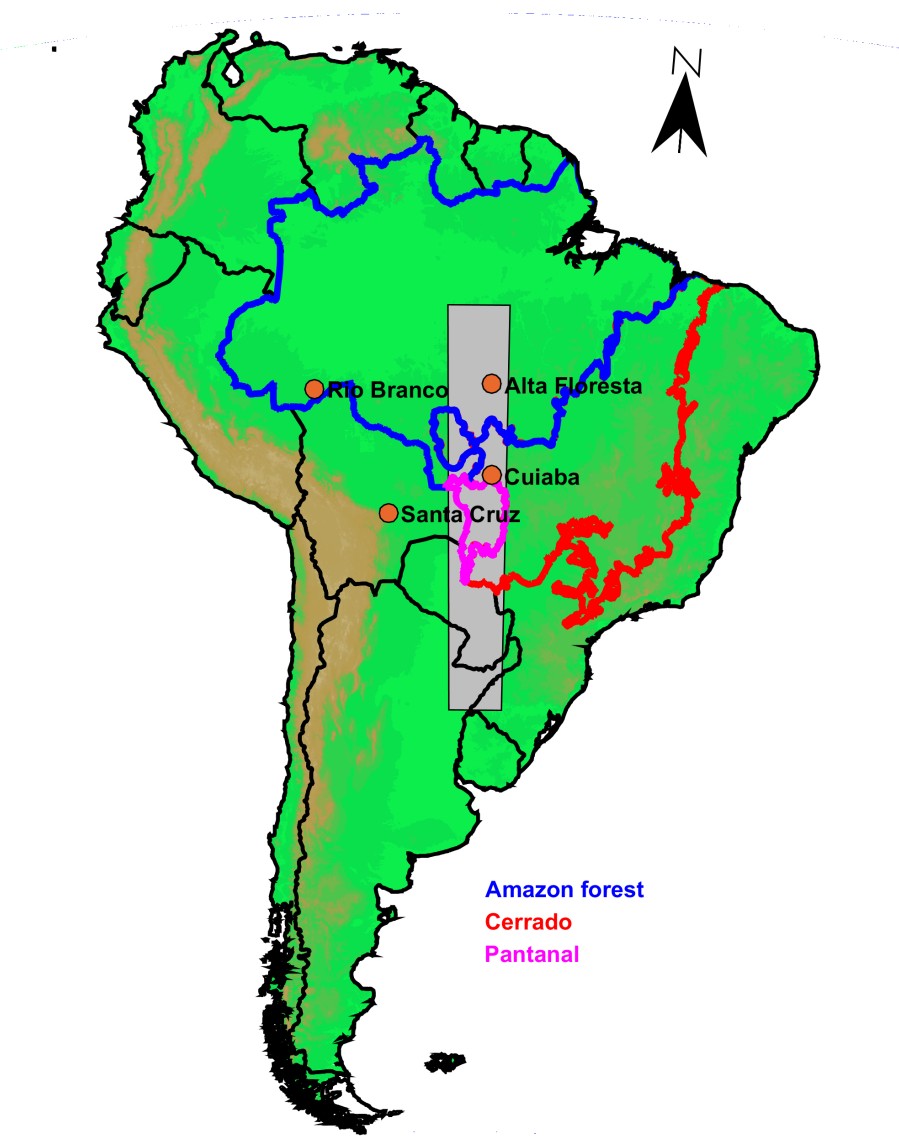

**Figure 1.** Spatial distribution of the Brazilian biomes of Amazon tropical forest, Cerrado and Pantanal. Locations of the AERONET stations considered in this study are also depicted(Solid brown circles). Grey shaded areas represent a transect defined to study north-south smoke loading variability and transport taking Pantanal west and east borders as reference.



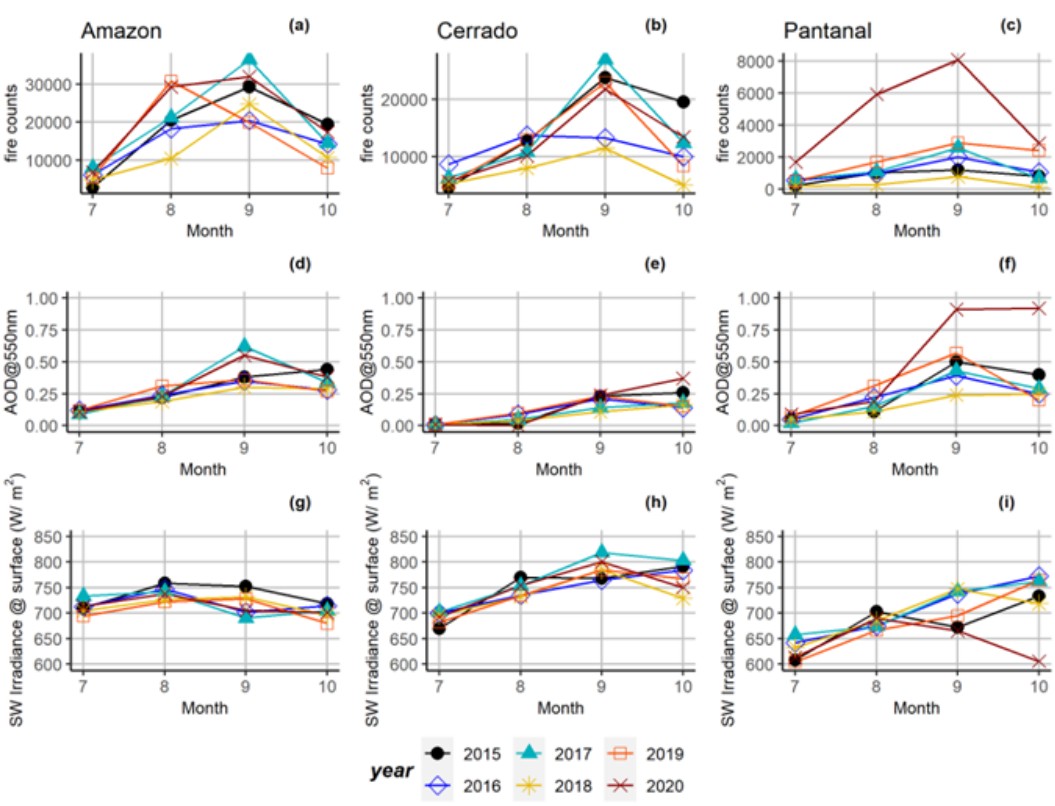

**Figure 2.** Monthly (from July to October) evolution of fire counts (a, b, c), aerosol optical depth (e, f, g) and downward solar irradiance at the surface (h, i, j) for the years 2015 to 2020 for different Brazilian biomes (Amazon, Cerrado and Pantanal).

**Figure 3.** Monthly mean values of AOD at 500 nm, single scattering albedo at 440 nm and aerosol radiative forcing at the surface from four AERONET sites: Alta Floresta, Cuiabá, Rio Branco and Santa Cruz.

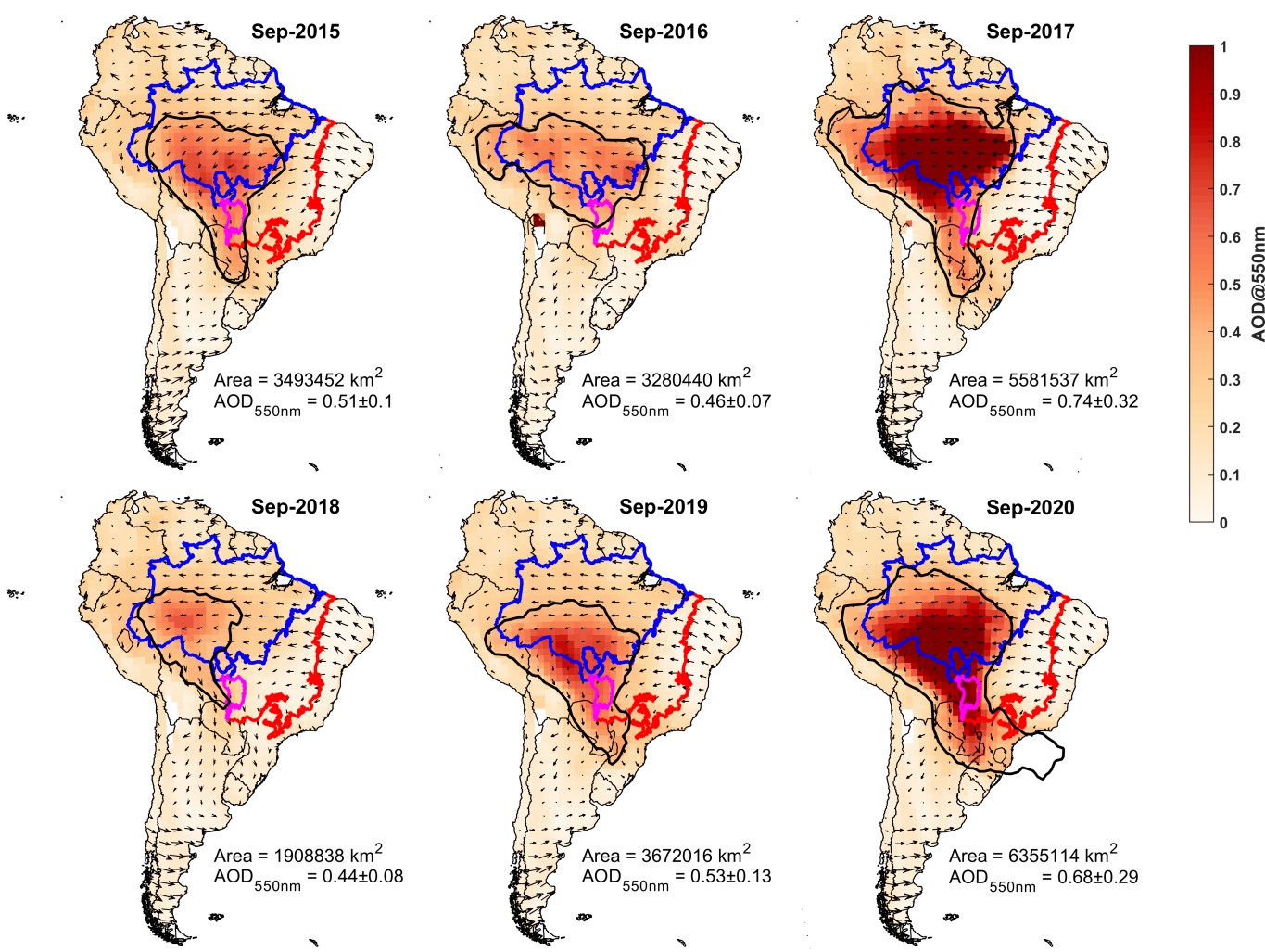

**Figure 4.** Spatial distribution of the mean aerosol optical depth at 550 nm (AOD550nm) and wind pattern at 850 hPa for September of different years. The area limited by the black solid line represents the regions with AOD550nm higher than 0.3, and its dimension (Area) and mean AOD550nm are also presented. The biomes' borders are represented by the colours blue (Amazon forest), red (Cerrado ecosystem) and magenta (Pantanal).



**Figure 5.** Spatial distribution of the mean surface solar irradiance under cloudless conditions and in the presence of aerosol for the month of September of different years. The biome's borders are represented by the colours blue (Amazon forest), red(Cerrado ecosystem) and magenta (Pantanal).



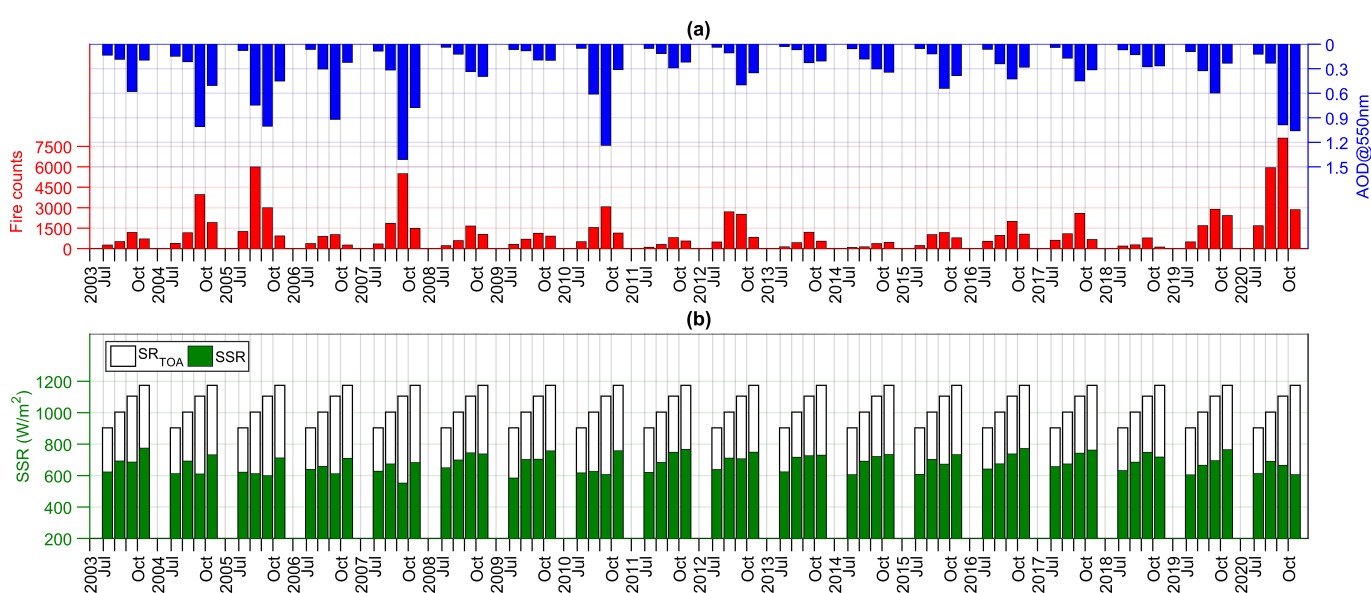

**Figure 6.** Pantanal biome interannual variability of (a) total fire counts, mean AOD@550nm and (b) and Top of the atmosphere (SRTOA) and Surface Solar Radiation (SSR) during the biomass burning season months (July, August, September and October).

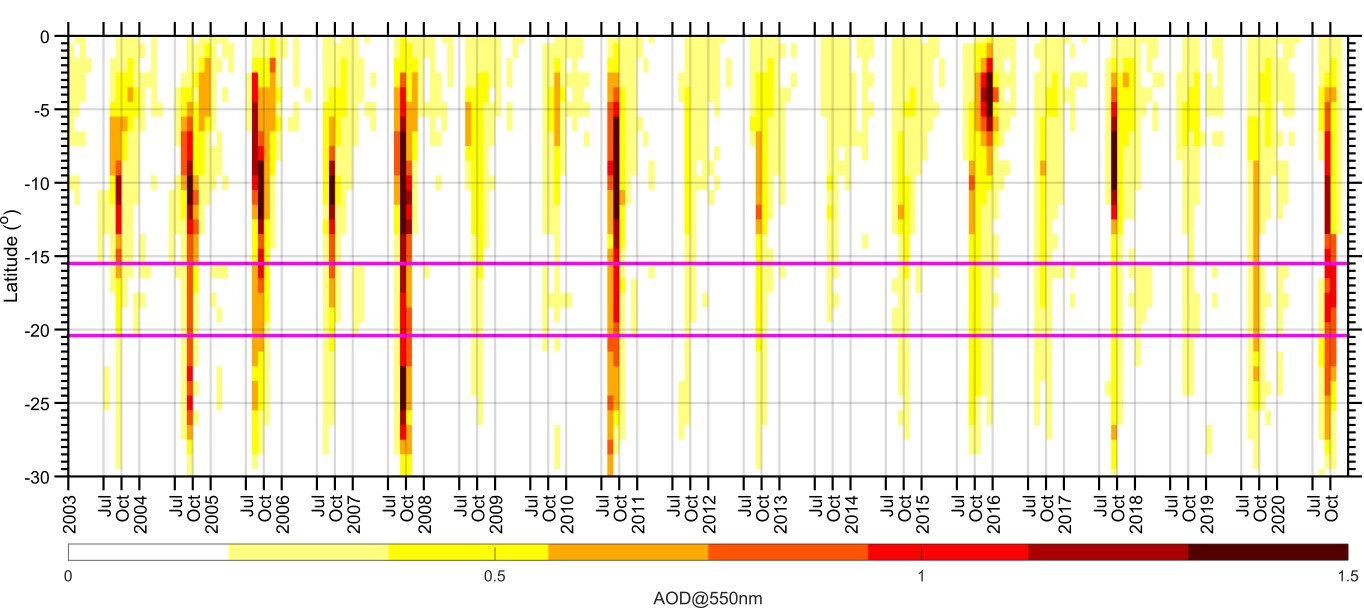

**Figure 7.** Hovmoller diagram of mean AOD at 550 nm from Amazon forest latitudes to Pantanal and downwind regions considering the average between Pantanal borders at west (58.5oW) and east (54.96oW) longitudes. The magenta line represents the north and southern borders of Pantanal. Each year is defined at its beginning, January first, while July and October positions are defined at the beginning of the month to highlight the biomass burning season.





**Figure 8.** Hovmoller diagram of the (a) zonal and (b) meridional wind speed at 850 hPa during the biomass burning season (July, August, September, October) considering the average between Pantanal borders at west (58.5oW) and east (54.96oW) longitudes. The magenta line represents the north and southern borders of Pantanal. Each year is defined at its beginning, January first, while July and October positions are defined at the beginning of the month to highlight the biomass burning season.





**Table 1.** Set of variables used in this study, their respective applications, sources and references.

| Variables | Application | Data sources | Reference |
|---|---|---|---|
| Aerosol Optical Depth at 500 nm (AERONET) and 550 nm (MODIS-Aqua) | Smoke plume loading and dimension | AERONET (Level, 1.5) https://aeronet.gsfc.nasa.gov/ MODIS Atmosphere Level 3 (L3) gridded products (Daily and Monthly) from the Aqua platform $(MYD08_D3/M3)$ $https://giovanni.gsfc.nasa.gov/giovanni/$ $Accessed on Feb 16, 2021.$ | Holben et al. (1998); King et al., 2013; Levy et al., 2013 |
| Aerosol Single Scattering Albedo | Smoke plume absorption efficiency | AERONET ( Level 1.5) https://aeronet.gsfc.nasa.gov/ | Dubovik and King (2000) |
| Aerosol radiative forcing | Radiative impact | AERONET (Level 1.5) https://aeronet.gsfc.nasa.gov/ | Garcia et al. (2012) |
| Wind components(zonal and meridional) at 850 hPa | Circulation pattern | MERRA 2- reanalysis atmospheric variables. https://disc.gsfc.nasa.gov/datasets/M2TMNXSLV_5.12.4 Accessed on November 11, 2021. | Gelaro, et al., 2017;GMAO (2015) |
| Fire counts | Fire Activity and Emission | http://queimadas.dgi.inpe.br/queimadas/portal-static/estatisticas$_e$stados/ − $Accessed on December 23, 2020.$ | Morisette et al. (2005) |

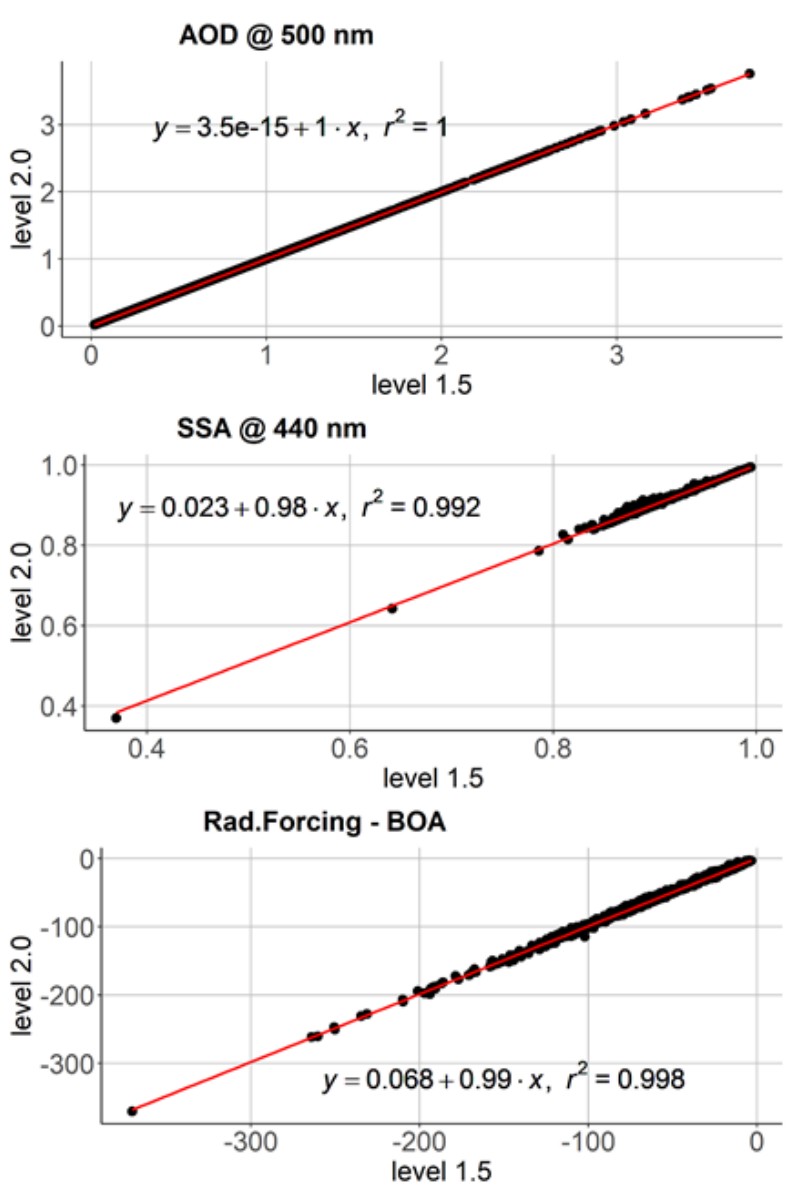

**Figure A1.** Scatterplot of level 2.0 versus level 1.5 for AOD at 500 nm, SSA at 440 nm and radiative forcing at the surface from AERONET.

Red line indicates the best line fit and the coefficients of the linear regression are also presented.