# Peer review of "South America 2020 Regional Smoke Plume: Intercomparison with 2 previous years, impact on solar radiation and the role of Pantanal biomass burning season"

_Atmospheric Chemistry and Physics, 2021_

## Referee Comment (RC2)

Comment on acp-2021-1086:

The paper has identified the irregularity of the biomass burning in the Amazon Forest for 2020 compared to the previous 6 years, which has involved the Pantanal biome. Evidence of the hotspot counts, AOD from MODIS and AERONET and retrieved solar irradiance have clearly shown the burning anomaly of the year. The statements are well discussed and proved but the contributing factors of the anomaly are not discussed in the paper.

Firstly, there is less specification on the main anomaly that is discussed whether it is a burning or emission or something else? Secondly, the factors that cause these anomalies should be understood, such as reason of burning (natural forest fire/human activities), source of burning (where and type of land cover burnt), weather condition (local or transported emission/conducive to sustain burning/conducive to fire spread). Since the year of 2020 is the COVID year, please do mention whether it played a role in the 2020 burning condition. A good background information would be helpful for the audience and further analysis.

In the result analysis (Section 3.2, 3.3), the analysis time frame of the data should be consistent. Hotspot, AOD and solar radiance data from 2003-2020 (Figure 6 – 8) were used for the chronological change of the burning events in Pantanal biome, but not for the spatial map (Figure 4 – 5) where the anomaly is identified. The inclusion of long-term dataset could assist on the understanding of spatial distribution of the burning condition more clearly. In the discussion section (Section 3.3) 2020 has been referred to the previous burning season in 2000s several times, in terms of the similarity of hotspot amount and AOD level. No detailed information is provided on the cause of the burning, even there are multiple similar occasions that have had happened in the past, which might not be so much of an anomaly but reoccurrence if longer period data (2003-2020) is considered.

Overall, the paper is not well structured, and information are clumped together in long paragraphs. Please split the lengthy paragraph or introduce sub-section for clarity. A major revision is required before the paper is deemed suitable for acceptance and publication.

Specific comments:

Line 56-60: Please provide the basic information on the size and the typical cause of burning in the different biomes. It would be helpful to elaborate the uniqueness of burning/emission condition in Pantanal.

Line 222-223: How about 2003?

Line 234-237: The role of weather anomaly is mentioned here but just for year of 2015-2016. How about the other year of extreme burning between 2003-2010 (Line 222-225)? More explanations need to be provided.

---

## Author Comment (AC1)

Atmos. Chem. Phys. Discuss., referee comment RC1
https://doi.org/10.5194/acp-2021-1086-RC1, 2022

[Figure]

**Comment on acp-2021-1086**

Lorraine Remer (Referee)

Referee comment on "South American regional smoke plume in recent years: main sources and impact on solar radiation focusing on the Pantanal 2020 biomass burning season" by Nilton Évora do Rosário et al., Atmos. Chem. Phys. Discuss., https://doi.org/10.5194/acp-2021-1086-RC1, 2022

We would like to thank Dr. Remer for the careful revision and thoughtful suggestions.

The replies to the questions, comments and suggestions are in blue color and the provided figures and references are displayed at the end of the current document. The figures presented here have been used to improve the manuscript's new version.

The authors present an observational analysis of intraseasonal and interannual characteristics of regional fire counts, smoke aerosol optical depth (AOD) and its radiative consequences for the South American dry season. The focus is on the unusual severity of the regional smoke pall in 2020 and the apparent significant increase of fire activity in a specific biome, the Pantanal. The authors cite news reports and agency news releases that capture the public's fascination with the event. Thus, the authors begin the study already with a qualitative understanding and expectation of results. However, there are questions that can only be answered with quantitative analysis.

■ How much of the **smoke anomaly** is due to anomalous fires in the Pantanal and how much is due to other factors?

R: Indeed, this is an important question. However, we are aware that to respond accordingly, it would be needed more than observational analysis. Certainly a modeling experiment would be required, either to explain the contribution of Pantanal anomalous fires to any anomaly in the regional smoke loading or to explain the role of other factors. Since an integrated observational-modeling study is out of the scope of the current manuscript, we tried to improve the manuscript by moving further in the discussion based on observational analysis and considering the valuable suggestion provided by both reviewers. While we did not provide a quantitative response to the posed question, we hope that with the new analysis and expected improvement we were able to add value to the manuscript discussion and analysis. Certainly, a modeling study guided by this relevant question raised by the reviewer is being considered as a continuity of this study.

At this point, as mentioned by the reviewer, clarity on the use of the term smoke anomaly is important, not just to specify when we are talking about the Smoke Over Pantanal (SOP) or about the Regional Smoke Plume (RSP), but also to be sure that indeed a smoke anomaly was the case (at least a significant one). After further analysis, adding to the comparison previous years marked by a strong regional smoke plume (ex. 2004, 2005, 2007, 2010), from the perspective of the regional smoke plumes, and considering the last two decades, 2020 did not stand as one of the topmost polluted years. The years 2004, 2007, 2010, 2017 presented stronger regional plumes (Figure 1). So, this is an important aspect that we decided to clarify in this revised version, so we can be as clear as possible about aspects of 2020 biomass burning season (BB) which could be considered as substantially different from previous years (anomalies). Regarding the regional smoke plume loading, if compared with the cited years (2004, 2005, 2007, 2010) and the climatology of the RSP, that was not the case, and neither was the fire count over the biomes Amazonia and Cerrado, as will be shown. Pantanal 2020 BB season was indeed the one that presented relevant deviations from historical features* (at least within the period analyzed by this manuscript (Figure 2) and according to INPE longest time series of fire count, which started in 1980's (Marengo et al, 2021). However, while the fire count in Pantanal in September 2020 was unprecedented within the monitored period, AOD was not. Only October 2020 presented AOD levels not seen in previous Octobers within the timeframe analyzed.

◾If due to other factors, how much is due to enhanced fire activity in other biomes and how much due to anomalous meteorology?

R: This question somehow is related to the previous one, and to provide a quantitative response it would also require some sort of modeling analysis. While we did not go in that direction, we've tried to include more observational analysis that could help to improve our consideration in this regard. Anomalous climate conditions have been recognized to play a role in fire activity in 2020, the years of 2019 and 2020 were characterized by the worst drought in 50 years in Pantanal (Marengo et al. 2021). The drought scenario was not restricted to the Pantanal domain, and yet the biome was the one that experienced a record of fire count. Amazonia and Cerrado biomes, traditional sources of smoke towards Pantanal (especially Amazonia), did not experience a similar level of enhancement in fire activity, in 2020.

Figure 3 shows that typically mean AOD over Pantanal strongly responds to the mean AOD over Amazonia, and it used to be similar or a fraction of mean AOD over Amazonia. In 2020 that was not the case, when the mean AOD value over Pantanal substantially surplused that over the Amazon, during both September and October. This aspect along with the enhancement of fire activity within the Pantanal domain were those that can be highlighted as significant or unprecedented when compared with the previous analyzed years and that can be assumed as anomalies.

◾Is the smoke anomalous only in terms of aerosol loading, or have intrinsic optical properties changed?

R: It is difficult to answer this question focusing on Pantanal, since there was not an operational AERONET station or other measurements of aerosol intrinsic properties within Pantanal in 2020 (as far as we are aware). However, it is possible to do some evaluation analyzing AERONET stations distributed across the regional smoke domain to get some insight on this matter, which we did, and is presented below. Focusing on the closest AERONET site to the Pantanal border, the Cuiaba site, we did not find a significant statistical difference between monthly mean SSA in 2020 and previous years (Figure 4). The same was observed for the other sites of AERONET analyzed.

■What effect does the anomalous smoke have on the radiative balance and what consequences does this have?

As discussed before, no statistically significant difference was observed in the data from the AERONET sites, either in the radiative balance. When analyzing the regional effect of the smoke layer on the radiative balance from CERES, 2007 presented the strongest attenuation of downward solar irradiance reaching the surface, much higher that 2020 regional smoke plume.

■How anomalous is this activity, not only in recent memory, but over scales spanning generations?

R: There was no need to go further into the past to show that, from a regional perspective and when compared with the mean scenario (within the period 2003-2020), 2020 biomass burning season could not be identified as an exceptional anomalous year, either based on fire count or smoke loading. As illustrated in Figure 1, the years 2004, 2007 and 2010 were more polluted and presented higher fire counts (Figure 2), regionally speaking (Amazonia+Cerrado+Pantanal).

However, when focusing specifically on the Pantanal domain, there are two informations that exceptionally differ from typical values (considering the period 2003-2020): the fire count observed in September and the mean AOD over the Pantanal biome during October (Figure 2). The high fire count over Pantanal in September 2020 has not been seen in previous years. As illustrated by the numbers cited before, in 2020, the fire count over Pantanal was 3.6 times higher than the mean climatological value.

■What caused the anomalous fire activity in the Pantanal?

R: Fire activity in Brazil biomes historically has a strong relationship with mankind intervention and there is a vast literature supporting this. Therefore, 2020 fire activity in Pantanal and the exceptional anomalous fire activity observed had, as usual, the mankind component. However, recent researches support that mankind traditional intervention was propelled by two distinct aspects:

a) A fire-prone environment (climate extreme, Marengo et al. 2021, Libonati et al., 2020). According to Marengo et al. (2021), the years of 2019 and 2020 were characterized by the worst drought in 50 years in Pantanal. The accumulated precipitation during the wet season of these years was between 50 and 60% less than normal.

b) An unfavorable governance (poor management and lax laws). According to Libonati et al. (2020), a combination of climate extremes, poor management and lax laws was behind Pantanal anomalous fire activity. Outdated environmental regulations, slashing of resources for environmental protection and climate actions in recent years certainly contributed to build-up the mentioned unfavorable governance.

One must also point out that the two concurrent aspects were not exclusive to Pantanal in 2020, and yet the fire count figures across Cerrado and Amazonia were not exceptionally far from those of recent years. Therefore, there are still open questions about the specific behavior of mankind intervention in Pantanal in 2020, a lack of studies of human causes and responses to fires in the Pantanal has been recognized as a challenge to a full comprehension of what happened (Libonati et al., 2020).

The authors present analysis that address most of these bullet points. There's a significant

paper in this work, but I have to say that they don't pull the analysis together in a way that clearly provides the answers. Because of that I will recommend Major Revisions.

I have no need to remain anonymous. This is Lorraine Remer writing.

Thank you very much Dr. Remer, for your time and careful review! We really appreciated that!

Points to address are as follows:

■What is meant by **smoke anomaly** in this study? (a) Is it the overall smoke loading over the entire continent? (b) Is it just the smoke over the Pantanal? (c) Is it the smoke over the population centers of the Brazilian southeastern coast? At times, while reading, I had the feeling that the authors meant it to be (a), and then (b) and then (c). All of these are interesting, but the authors need to clarify when they are considering each one.

Let's assume that the main point is (b) because that is what Figure 6 addresses, although Figure 7 is more tuned to (a). Then I'm going to ask, "Why?". Why do we care specifically about how much smoke is over the Pantanal? It is a very small area from the regional perspective. I would think the question of "how much do Pantanal fires contribute to the regional big picture" to be the more interesting question rather than, "is the smoke above the Pantanal due to local or transported smoke". I mean, both questions are interesting, but the big picture is the bigger picture. If the authors find smoke and its consequences directly over the Pantanal to be the primary question to address, then they need to introduce the reason for this in the introduction… "The Pantanal represents a unique island of biodiversity in the region and smoke hanging over this area for up to six weeks has the potential for diminishing surface shortwave flux, stopping photosynthesis, interfering with primary productivity that has consequences as it cascades up the ecosystem." Or something like that. I know that that this is touched on here and there, but the paper needs to be structured in a way that makes this the primary focus.

If the authors are indeed looking more at the big regional picture (a) then there needs to be analysis presented " XX% of regional smoke is produced by the Patanal, representing only yy% of the regional surface area."

Or something like this. Or the authors could go in both directions. The paper is short. It could support two specific sections, one addressing (a) and one addressing (b). I don't need it to do both. I just need some clarity and focus communicated.

R: Although one could focus on one or another item, and indeed we are slightly tuned to Pantanal, the current version helps us to understand the need and importance of contextualizing both: (a)  the 2020 overall smoke loading over the continent. (b) the 2020 smoke over the Pantanal. Thus, we show that regionally 2020 was not a particular year from the perspective of the plume dimension, loading and optical properties, and  we explore what happened to Pantanal in 2020. In this sense, the focus on Pantanal presented more interesting results and analysis. And we adjusted the manuscript to highlight the importance of Pantanal (locally and regionally) to support our focus and to show how protected areas (as indigenous and reserves) were atypically burned in the biome in 2020, as shown in Figure 3 and further discussion ahead.

2.      Is there any insight gained from direct scatter plots of smoke vs. fire counts, and SW flux vs. AOD? Scatter plots of monthly means, for example, taking the points shown in Figure 2 and just throwing them into scatter plots. 4 month x 6 years. That's a 24 point scatter plot. The more fire counts, the more AOD, right? But if the Pantanal is more affected by advection than by local fires, there won't be much correlation. And maybe 2020 stands out, as an outlier. I don't know. It's just that right now the only thing I gain from Figure 2 is that 2020 is weird for both fire counts and smoke in the

Pantanal, but that smoke weirdness lags fire weirdness by one month. There are a lot of words describing this figure, but few of those words point to the focus of the study.

**R:** The scatter plots of smoke vs fire counts over Pantanal and of smoke over Amazonia vs smoke over Pantanal indeed provided interesting insights (Figure 3). Smoke vs fire counts over Pantanal shows that, in general, the higher the fire count is, the higher the AOD, but there are several exceptions (ex. August 2020). The scatter plot smoke over Amazonia vs smoke over Pantanal  shows that the smoke over Pantanal has a stronger relationship with smoke over Amazon than with fire counts within the biome, suggesting that AOD over Pantanal is more affected by advection than by local fires. However, 2020 does stand out as an outlier. Typically, mean AOD over Pantanal domain is similar or a fraction of that over Amazonia domain. That was not the case for September and October of 2020, when mean AOD over Pantanal was much higher than over Amazonia, an indication that local smoke played an atypical role to the smoke level over Pantanal.

Regarding the one month lag between fire counts and AOD values, we added an analysis of fire count distribution on top of the vegetation index (EVI) within the Pantanal domain to help clarify a hypothesis that these observational data can point out (Figure 3). From plots (a), August was characterized by high fire count, but relatively low AOD, while October by relatively low fire count and high AOD. The high AOD over Pantanal in October, despite the reduction in fire count, hardly is explained by advection from Amazonia. As can be seen in the plot (b), there was a reduction in smoke loading over Amazonia during October, and the regional map of AOD for October (Figure 1) shows that there was a spot of high AOD centered and over the Pantanal domain. Therefore, a possible explanation for the lag between fire count and AOD from August to September/October could be the nature of the material being burned within Pantanal. As the maps show, during August a reduced number of fires was within conservation and Indigenous areas (where higher biomass density is present). However, from September on, there was a significant increase of fire number within these areas, which could explain the larger aerosol emissions and, consequently, the increase in AOD.

3.    Figure 3 is interesting because of the SSA, but the question I need answered is too hard to find in these plots. Is the SSA different in 2020 or not? Will radiative effects only be controlled by loading, or do changing optical properties play a role? Any thought of trying some 24-point scatter plots here also?

**R:** In order to answer the question, we replaced Figure 3 of the manuscript first version to a new one( Figure 5 of the current doc), following the reviewer's suggestion. Instantaneous aerosol radiative forcing estimates from AERONET were plotted versus AOD at 550 nm, color coded by single scattering albedo (SSA) values, with data from July to October of the years 2003 to 2020. Data from 2020 are highlighted with red symbols. It is possible to observe that not only AOD but also SSA affect the downward solar irradiance at the surface and that no difference can be noticed in 2020 data. To complement the information, boxplot of  SSA at 440 nm for different sites and years are now presented in Figure 4.   From the boxplots, we conclude that SSA values from 2020 were similar to previous years.

4.    I thought Figure 4 was the most informative of the basic plots. Here you see the difference from year-to-year much better than in Figures 2 and 3. In 2020, the Pantanal stands out quite a bit darker than its immediate surroundings. This is the first place that I considered that local smoke might dominate AOD over the Pantanal. The authors also rightly point out the difference in flow between 2017 and 2020 that explains why the population centers of the southeastern coast were spared in 2017.

**R:** Despite the recognition of the informative value of this plot, following the new arrangement aiming to better contextualize the 2020 regional smoke plume and the smoke loading over Pantanal, the previous plot (Figure 4 of the first version submitted

and mention in the reviewer question) was replaced by a similar analysis (Figure 1 of the current doc) focusing on the intercomparison of 2020 regional smoke with the most polluted years (2004, 2005, 2007 and 2010) within the timeframe analyzed (2003-2020). The intercomparison now also includes maps from the month of October to highlight the high level of smoke restricted to the Pantanal domain and surroundings, which is evaluated as the most significant deviation from the historical perspective here analyzed (2003-2020), and in the context of Pantanal biome(Figure 2). In September 2020, despite the exceptional fire count level, the smoke loading over the Pantanal biome was not unprecedented. September of 2007 and 2010, for instance, presented AOD over Pantanal much higher than September 2020.

Related to the flow of the regional smoke plume towards the population centers of the southeastern coast, the new plot evidenced that this has also been seen in the past. For example, in 2004 and 2005 the monthly flow patterns were also towards the highly populated centers in the southeast of Brazil(Figure 1).

5.      Figure 5. Have the authors considered an anomaly plot instead of absolute irradiance? Interannual differences are hard to see now.

**R:** Following the reviewer's suggestion, absolute irradiance was replaced by anomaly considering the period from 2003 to 2020 as reference to define the month climatology (Figure 6). However, as described and justified in the reply to reviewer query number 4, the focus has been changed to the intercomparison between 2020 and the most polluted regional smoke plume (2004, 2005, 2007 and 2010) in the decades. We expect that now it is much clearer to highlight the effect of 2020 regional smoke plume on surface solar radiation (SSR), how  it stands when compared with the top polluted biomass burning season in the timeframe analyzed (2003-2020). September of 2020 anomalies in SSR are comparable to some of the most polluted years, but September 2007 presented the highest anomalies. Considering the atypical amount of smoke over Pantanal during october of 2020, we decided to include october in the comparison.

Figure 6. Another good plot. It was very important to do the longer time series. Knowing that 2020 still had unprecedented fire counts in smoke, we see that the anomaly isn't that much greater than in the 2000s. People's memories are short. They forget how bad things were in the previous generation. Although, I didn't see anything interesting in the SW flux plot and I recommend dropping it. Also, shouldn't there be fire counts that go back another decade? It's not necessary, but it would be interesting. Fire counts were intense in the late 1980s and 1990s.

**R:** Pantanal 2020 BB season was indeed the one presenting relevant deviations from historical features, at least within the  analyzed period in this manuscript (Figure 2) and according to INPE longest time series of fire count, which started in 1980's, as pointed out by Marengo et al. (2021). We also decided to include the same time series plots for Amazonia and Cerrado biomes. So one can have a better overview of the comparison between the biomes.

6.      Figure 7. I like this one also. This is a very nice summary of the entire situation over time.

**R:** Thank you, we kept this summary.

7.      Figure 8. I couldn't understand this at all. I couldn't find in it the things mentioned in the text.  Maybe it is just me.

**R:** This plot aimed to explore the role of circulation interannual variability, via wind

components. However, given that it did not present clear information, and considering that more new plots were added to the analysis, we understand that this plot may be excluded to make space for further discussion required by the new added plots.

8.      The three biomes, Amazonia, Cerrado and Patanal are mentioned very early in the introduction, before the map, sort of with the expectation that the reader already knows what they are. I suggest describing each one, as soon as they are mentioned. One thing that I struggled with is the relative sizes of these biomes. Patanal is so tiny. Why does anybody care?   Also, Caatinga, Pampas, Mata Atlantica are mentioned with the expectation that the reader knows what these are, and we don't.

**R:** The description and location of the biomes are reorganized, including Caatinga, Pampas and Mata Atlantica, so the reader is better contextualized. A new map was developed (Figure 7) to clearly depict these biomes. Indeed, Pantanal size is much smaller than the other biomes, however it's unique biodiversity and its role to the regional hydrological cycle are of great relevance.

9.      Lines 85-90. We need more information on the surface irradiance. What is meant by cloud free in terms of footprint size? What is used for aerosol model to make the calculations? If there were to be a change in SSA, would the irradiances that are analyzed reflect that information?

**R:** Clouds and aerosol properties are obtained from MODIS with 10km nadir resolution and then projected (with the corresponding weights) into CERES SSF Level 2 footprint of 20 km (point-spread-functions are used to convolve higher resolution data into CERES 20km optical footprint). Finally, to obtain the SSF Level 3 product, the results are interpolated into 1º×1º latitude/longitude grid size. In this work a cloud-free scene is considered a scene that has aerosol retrievals on a given grid cell. According to CERES SSF 1 degree documentation, clear-sky parameters are only computed if cloud fraction is less than 0.1% in a given footprint.

(https://ceres.larc.nasa.gov/documents/DPC/DPC_current/pdfs/DPC_SSF1deg-Day_R5V1.pdf).

The aerosol model used in CERES surface irradiance calculations is very briefly described in Kratz, 2020. It uses near-real time daily AODs from the Model for Atmospheric Transport and Chemistry (MATCH). The Optical Properties of Aerosols and Clouds (OPAC) Global Aerosol Data Set (GADS) was used to obtain the intrinsic aerosol properties, such as single scattering albedo and asymmetry parameter (Hess et al., 1998).

10. Line 92. What is meant by Aqua bouncing?

**R:** We believe there was a mistranslation here. What actually happened was a problem with the data formatter for the solid state recorder aboard the Aqua satellite that resulted in a data acquisition gap from August 16 through September 2, 2020 for MODIS Aqua products. This information has been corrected in the text and the references on this issue were included.

Refs.: https://lpdaac.usgs.gov/news/aqua-satellite-anomaly/

https://landweb.modaps.eosdis.nasa.gov/cgi-bin/NRT/displayCase.cgi?esdt=MYD&caseNum=PM_MYD_20229_NRT&caseLocation=cases_data

11.      Lines 244-247. These are some of the most important statements in the manuscript. Somehow the paper should be structured to get there. Also, this is where I started wondering about a scatter plot.

**R:** We added new plots and performed a reorganization in the discussions to accomplish this.

12.     Lines 252-253. Suddenly the authors start using the word "emissions". There are no emissions analyzed. The authors are working with fire counts, which are not the same as emissions. Then they state that the contribution of emissions from the Pantanal rivals that from Amazonia and Cerrado. In absolute terms Figure 2 shows that the total fire counts from the Pantanal never reaches the total numbers of the other biomes (because of its small area).  Where else do the authors find evidence to support that statement?

**R:** The reviewer is right, the word emissions should not be used here, and there is no evidence to suggest that emission from Pantanal's rivals that of Amazonia and Cerrado in 2020. What can be suggested or inferred, based on fire counts, is that the relative contribution of Pantanal's emission to the regional smoke plume in 2020 was higher than the other analyzed years, given that the high amount of smoke over the biome in 2020 was associated with an explosion in local fire, especially during September and October.

13.     I also note that a lot of statements found in Section 3.3 should really go into the Conclusions, especially Lines 244 and beyond.

**R:** The indicated statements were moved to the Conclusions.

14.     The Conclusions are mostly strong and well-stated. There's one sentence in the Conclusions that was not supported in the body of the manuscript. "For Pantanal, 2020 was a very particular year, not exactly due to the aerosol loading over the biome, since in September of 2007 the biome experienced a higher monthly mean AOD, but due to the contribution of the local fire emission to the regional smoke plume. "

**R:** With the new analysis performed based on scatter plots of fire count vs smoke over Pantanal and AOD_Amazonia vs AOD_Pantanal (Figure 3), we understood that now we are able to support this adjusted sentence:  *"For Pantanal, 2020 was a very particular year, not exactly due to the aerosol loading over the biome, since in  September of 2007 the biome experienced a higher monthly mean AOD, but due to the contribution of local fire emission to the local smoke plume, which surpassed advection from Amazonia biomass burning areas"*.

There is no analysis that states that the Pantanal contributes XX% of the smoke in the regional smoke plume. Even after all of this analysis, this statement is still based on qualitative discussion. What has this study contributed to supporting this statement that the NYT, BBC and Le Monde have not?

R: Yes, that's right, current analysis is not able to quantify Pantanal contributions to the regional smoke plume. With new observational plots and analysis we hope that the manuscript is providing a better and more consistent discussion.

15.     Finally, and the most frustrating… Why did the Pantanal burn in 2020? What is fundamentally different about this year that fire erupted in a way unseen for over a decade? Drought? Human intervention or non-intervention? Why? While the paper can be published without answering this question, it is the question in the reader's mind as they read through the analysis. Why the fire eruption in the Pantanal in 2020? There's an elephant in the room that nobody is mentioning. A paragraph in the Conclusions with some educated hypotheses in a Discussion format would be satisfying.

R: As discussed above, the most reliable hypothesis about 2020's Pantanal fires is that it was due to a combination of a drought-prone environment and lack of effective public policies to protect the environment against man-made fires. Actually, unfortunately there

was a huge throwback on the steps previously taken regarding environmental laws, additionally resources for environmental protection and climate actions have been slashed. During the last years, particularly after the 2018 elections, Brazilian authorities have adopted against-the-environment and pro-agricultural-livestock-expansion speeches. Under the new government, Brazil's main environmental enforcement agency, Ibama, is weakened and it reduced the number of fines for those involved with deforestation. In this context, indigenous and conservation areas have become exposed targets for those aiming to expand illegally agricultural land.

[Figure]

Figure 1: Regional Smoke Plume (a) September ; (b)October

[Figure]

Figure 2: Fire count and AOD@550 nm interannual variability (a) Pantanal; (b) Amazonia; (c) Cerrado

[Figure]

Figure 3: (a) AOD@550 nm Pantanal versus Fire count Pantanal; (b) AOD@550 nm Pantanal versus AOD@550 nm Amazonia; (c) Fire spots distribution across Pantanal for August, September and October of 2020 using Enhanced Vegetation Index(EVI) as background and highlighting Indigineous and conservation areas; (d) Pantanal total fire count and fire count within Conservation and Indigenous areas.

[Figure]

Figure 4: Interannual variability of mean Single Scattering Albedo (SSA) during biomass burning season (Aug-Sept-Oct) from AERONET stations (Please, refer to Figure 7 to see the site geographical distribution).

[Figure]

Figure 5 - Aerosol direct instantaneous Radiative forcing as function of AOD at 550 nm at AERONET sites highlighting 2020 (red) against historical value (2003 – 2019)

[Figure]

Figure 6 - Surface Solar Radiance anomaly: (a) September; b) October

[Figure]

Figure 7 -  New map representing Brazil's biomes and  AERONET sites.

**References:**

LIBONATI R ET AL. 2020. Rescue Brazil's burning Pantanal wetlands. Nature 588: 217-219. https://doi.org/10.1038/d41586-020-03464-1.

Marengo, J. A. et al. Extreme drought in the Brazilian Pantanal in 2019–2020: Characterization, causes and impacts. Front. Water3, 639204 (2021).

---

## Author Comment (AC2)

We would like to thank the reviewer for his/her important and meaningful questions. Below(in blue) are the replies to the comments and questions. The referred figures and references are at the end of this doc.

Comment on acp-2021-1086:

The paper has identified the irregularity of the biomass burning in the Amazon Forest for 2020 compared to the previous 6 years, which has involved the Pantanal biome. Evidence of the hotspot counts, AOD from MODIS and AERONET and retrieved solar irradiance have clearly shown the burning anomaly of the year. The statements are well discussed and proved but the contributing factors of the anomaly are not discussed in the paper.

Firstly, there is less specification on the main anomaly that is discussed whether it is a burning or emission or something else? Secondly, the factors that cause these anomalies should be understood, such as reason of burning (natural forest fire/human activities), source of burning (where and type of land cover burnt), weather condition (local or transported emission/conducive to sustain burning/conducive to fire spread). Since the year of 2020 is the COVID year, please do mention whether it played a role in the 2020 burning condition. A good background information would be helpful for the audience and further analysis.

**R:** First, we would like to clarify that, following both reviewers comments, we reassess the identification of anomalies in the context of Pantanal and Brazil 2020 biomass burning season. We change the time frame of the analysis, looking also for the other biomes (Amazonia and Cerrado, not just Pantanal), for the period between 2003-2020.

Looking further into the past, it shows that, from a regional perspective and when compared with the mean scenario (within the period 2003-2020), 2020 biomass burning season could not be identified as an unprecedented and exceptional anomalous year, either based on fire count or smoke loading. As illustrated in Figure 1, the years 2004, 2007 and 2010 were more polluted and presented higher fire counts (Figure 2), regionally speaking (Amazonia+Cerrado+Pantanal). Therefore, regionally, the focus has been changed to the intercomparison between 2020 and the most polluted regional smoke plume (2004, 2005, 2007 and 2010) occurred between 2003 and 2020.

However, when focusing specifically on the Pantanal domain, there are two aspects evidencing that 2020 exceptionally differs from typical years (considering the period 2003-2020): the fire count observed in September and the mean AOD over the Pantanal biome during October (Figure 2). The high fire count over Pantanal in September 2020 has not been seen in previous years. Since there is already a detailed study on the climate conditions and its potential role on the Pantanal 2020 Biomass burning season (Marengo et al. 2021) we try to better understand the relationship between smoke levels over Pantanal and local fire count and smoke in the Amazon, from where transport to Pantanal is important (Figure 3).

To contextualize the factors associated with Pantanal exceptional fire count in 2020, as background information for the new version of the manuscript, we cited Marengo et al. (2021) and Libonati et al. (2020) studies as follow:

Fire activity in Brazil biomes historically has a strong relationship with mankind intervention and there is a vast literature supporting this. Therefore, 2020 fire activity in Pantanal and the exceptional anomalous fire activity observed had, as usual, the mankind component. However, recent researches support that mankind traditional intervention was propelled by two distinct

aspects:

a) A fire-prone environment (Marengo et al. 2021, Libonati et al., 2020). According to Marengo et al. (2021), the years of 2019 and 2020 were characterized by the worst drought in 50 years in Pantanal. The accumulated precipitation during the wet season of these years was between 50 and 60% less than normal.

b) An unfavorable governance (poor management and lax laws). According to Libonati et al. (2020), a combination of climate extremes, poor management and lax laws was behind Pantanal anomalous fire activity. Outdated environmental regulations, slashing of resources for environmental protection and climate actions in recent years certainly contributed to build-up the mentioned unfavorable governance.

One must also point out that the two concurrent aspects were not exclusive to Pantanal in 2020, and yet the fire count figures across Cerrado and Amazonia were not exceptionally far from those of recent years. Therefore, there are still open questions about the specific behavior of mankind intervention in Pantanal in 2020, a lack of studies of human causes and responses to fires in the Pantanal has been recognized as a challenge to a full comprehension of what happened (Libonati et al., 2020).

In a recent study (Vale et al., 2021), the conclusion was that the current administration took advantage of the COVID-19 pandemic to intensify a pattern of weakening environmental protection in Brazil. The study examined the effects of the pandemic on environmental protection and legislation in Brazil in the current administration and showed 57 legislative acts aimed at weakening environmental protection, almost half of which in the seven-month period of the pandemic in Brazil. The study also found a 72% reduction in environmental fines during the pandemic, despite the increase observed in Amazonian deforestation during the analyzed period. It is important to stress that this context encouraged people to set fire around the country, particularly in indigenous and protected areas. This background is being considered in the new version of the manuscript.

In the result analysis (Section 3.2, 3.3), the analysis time frame of the data should be consistent. Hotspot, AOD and solar radiance data from 2003-2020 (Figure 6 – 8) were used for the chronological change of the burning events in Pantanal biome, but not for the spatial map (Figure 4 – 5) where the anomaly is identified. The inclusion of long-term dataset could assist on the understanding of spatial distribution of the burning condition more clearly.

R: In the revised version of the manuscript, we kept the time series plots of fire count and AOD (Figure 2) from 2003 to 2020, and we extended it to Cerrado and Amazonia biome, but the time frame was changed for the spatial maps. We included maps for September and October of the more recent years (between 2003 to 2020) identified as the most polluted in terms of the regional smoke plume(Figure 1) to be compared to the results for 2020 (2004, 2005, 2007, 2010, 2017). So, instead of plotting all the maps of September from 2003 to 2020, we focus on the most polluted years and add maps for October to highlight the atypical high level of smoke over Pantanal at the end of the bb season. We did it for both, AOD (Figure 1) and Surface Solar Radiance anomaly (Figure 4). The remaining years, from the previous time frame, would be added as supplementary material.

In the discussion section (Section 3.3) 2020 has been referred to the previous burning season in 2000s several times, in terms of the similarity of hotspot amount and AOD level. No detailed information is

provided on the cause of the burning, even there are multiple similar occasions that have had happened in the past, which might not be so much of an anomaly but reoccurrence if longer period data (2003-2020) is considered.

R: The reviewer is corrected and the revised version of the manuscript tries to make it clearer what the authors considered anomalous in 2020. In fact, in terms of the regional smoke plume, in 2007 higher AOD values were detected and covered a larger area in South America. As mentioned previously, in Pantanal, however, in 2020, the fire count detected in September and the mean AOD over the Pantanal biome during October differ from previous years values.  So, regionally (Brazil), within the time frame analyzed, 2020 was not different from past bb season (Figure 1) in terms of smoke loading and total fire count, it was Pantanal that present aspects not seen  in the previous years, as mentioned.

Overall, the paper is not well structured, and information are clumped together in long paragraphs. Please split the lengthy paragraph or introduce sub-section for clarity. A major revision is required before the paper is deemed suitable for acceptance and publication.

R: We reformulate the structure of the paper, we expect that the revised version is better structured, making the readability easier.

Specific comments:

Line 56-60: Please provide the basic information on the size and the typical cause of burning in the different biomes. It would be helpful to elaborate the uniqueness of burning/emission condition in Pantanal.

R: In general, slash and burn practices are tools in the deforestation process of the original vegetation. Fires are also set to clear the land from residues from the previous crop season. Particularly in 2020, analyzing fire count map over EVI map, mankind intervention over indigenous and protected areas was clearly identified (Figure 3). The social and economic motivations, specially during 2020 season, need to be better studied but they are out of the scope of the present work, although as pointed before, the current administration's efforts to weaken environmental protection in Brazil (Vale et al., 2021), lax laws and lack of fiscalization must have contributed to this scenario (Libonati et al, 2020).

Line 222-223: How about 2003?

R:  Yes, 2003 could be included since it is the beginning of the time series, nevertheless, the fire count and AOD over Pantanal in 2003 were much lower than those identified as polluted. This has been adjusted in the new version of the manuscript.

Line 234-237: The role of weather anomaly is mentioned here but just for year of 2015-2016. How about the other year of extreme burning between 2003-2010 (Line 222-225)? More explanations need to be provided.

R: Considering the new arrangement, to focus the spatial comparison between 2020 and the most polluted years between 2003 and 2020, what we found in common among the polluted years(including 2020) is the prevalence of drought conditions over the center-west region of Brazil and southern Amazonia (Figure 5).  As in 2020, the drought conditions also spread across Pantanal in 2010. This again points out  to the particularity of the human component  in the explosion of fire in september of 2020. This has been addressed in the new version of the manuscript.

(a)

[Figure]

(b)

Figure 1: Regional Smoke Plume (a) September ; (b)October

[Figure]

Figure 2: Fire count and AOD@550 nm interannual variability (a) Pantanal; (b) Amazonia; (c) Cerrado

[Figure]

Figure 3: (a) AOD@550 nm Pantanal versus Fire count Pantanal; (b) AOD@550 nm Pantanal versus AOD@550 nm Amazonia; (c) Fire spots distribution across Pantanal for August, September and October of 2020 using Enhanced Vegetation Index(EVI) as background and highlighting Indigineous and conservation areas; (d) Pantanal total fire count and fire count within Conservation and Indigenous areas.

[Figure]

[Figure]

[Figure]

Figure 4 - Surface Solar Radiance anomaly: (a) September; b) October

[Figure]

Figure 5 - Brazil Standard Precipitation Index (SPI) in September for the 3 month scale and for the years 2004, 2005, 2007, 2010, 2017 and 2020. Red areas represent extremely dry conditions
(source: http://clima1.cptec.inpe.br/spi/pt)

**References:**

LIBONATI R ET AL. 2020. Rescue Brazil's burning Pantanal wetlands. Nature 588: 217-219. https://doi.org/10.1038/d41586-020-03464-1.

Marengo, J. A. et al. Extreme drought in the Brazilian Pantanal in 2019–2020: Characterization, causes and impacts. Front. Water3, 639204 (2021).

Vale, M. M., Berenguer, E., Argollo de Menezes, M., Viveiros de Castro, E. B., Pugliese de Siqueira, L., & Portela, R. (2021). The COVID-19 pandemic as an opportunity to weaken environmental protection in Brazil. Biological

conservation, 255, 108994. https://doi.org/10.1016/j.biocon.2021.108994

---

## Author Response (AR2)

To attend the reviewers' comments and suggestions we did substantial modifications in the text, including in the title. We hope that the manuscript is now clearer and mode detailed in its purpose to describe the objectives, methods and results.

Due to the substantial differences between the two versions of the manuscript, the track changes version was too polluted. Therefore, we added a track change version indicating the major adjustments performed, and in the current response file to the reviewers we highlighted the specific places and plots in the track changes version where we try to responde each question posed by the reviewers.

**Black - Reviewer questions**

**Blue - Authors responses**

**The response to the two reviewers are displayed in this document**

**RESPONSE TO REVIEWER 01**

We would like to thank Dr. Remer for the careful revision and thoughtful suggestions. The replies to the questions, comments and suggestions are in blue color and the provided figures and references are displayed at the end of the current document. The figures presented here have been used to improve the manuscript's new version. The place and references of the changes in the new manuscript version to respond to the reviewer are marked in yellow.

The authors present an observational analysis of intraseasonal and interannual characteristics of regional fire counts, smoke aerosol optical depth (AOD) and its radiative consequences for the South American dry season. The focus is on the unusual severity of the regional smoke pall in 2020 and the apparent significant increase of fire activity in a specific biome, the Pantanal. The authors cite news reports and agency news releases that capture the public's fascination with the event. Thus, the authors begin the study already with a qualitative understanding and expectation of results. However, there are questions that can only be answered with quantitative analysis.

■How much of the **smoke anomaly** is due to anomalous fires in the Pantanal and how much is due to other factors?

R: Indeed, this is an important question. However, we are aware that to respond accordingly, it would be needed more than observational analysis. Certainly a modeling experiment would be required, either to explain the contribution of Pantanal anomalous fires to any anomaly in the regional smoke loading or to explain the role of other factors. Since an integrated observational-modeling study is out of the scope of the current manuscript, we tried to improve the manuscript by moving further in the discussion based on observational analysis and considering the valuable suggestion provided by both reviewers. While we did not provide a quantitative response to the posed question, we hope that with the new analysis and expected improvement we were able to add value to the manuscript discussion and analysis. Certainly, a modeling study guided by this relevant question raised by the reviewer is being considered as a continuity of this study.

At this point, as mentioned by the reviewer, clarity on the use of the term smoke anomaly is important, not just to specify when we are talking about the Smoke Over Pantanal (SOP) or about the Regional Smoke Plume (RSP), but also to be sure that indeed a smoke anomaly was the case (at least a significant one). After further analysis, adding to the comparison previous years marked by a strong regional smoke plume (ex. 2004, 2005, 2007, 2010), from the perspective of the regional smoke plumes, and considering the last two decades, 2020 did not stand as one of the topmost polluted years. The years 2004, 2007, 2010, 2017 presented stronger regional plumes (Figure 1 below, which was incorporated as Figure 7 at page 16 of the new version of the manuscript). So, this is an important aspect that we decided to clarify in this revised version, so we can be as clear as possible about aspects of 2020 biomass burning season (BB) which could be considered as substantially different from previous years (anomalies). Regarding the regional smoke plume loading, if compared with the cited years (2004, 2005, 2007, 2010) and the climatology of the RSP, that was not the case, and neither was the fire count over the biomes Amazonia and Cerrado, as will be shown. Pantanal 2020 BB season was indeed the one that presented relevant deviations from historical features* (at least within the period analyzed by this manuscript (Figure 2, below and at page 17 of the revised version of the manuscript) and according to INPE longest time series of fire count, which started in 1980's (Marengo et al, 2021). However, while the fire count in Pantanal in September 2020 was unprecedented within the monitored period, AOD was not. Only October 2020 presented AOD levels not seen in previous Octobers within the timeframe analyzed. The modified phrases with the above explanations can be found at page 2, lines 56 to 70, section 3.1, pages 5 and 6 and section 3.2, page 8.

■If due to other factors, how much is due to enhanced fire activity in other biomes and how much due to anomalous meteorology? (Pag. 6 174 - 177)

R: This question somehow is related to the previous one, and to provide a quantitative response it would also require some sort of modeling analysis. While we did not go in that direction, we've tried to include more observational analysis that could help to improve our consideration in this regard. Anomalous climate conditions have been recognized to play a role in fire activity in 2020, the years of 2019 and 2020 were characterized by the worst drought in 50 years in Pantanal (Marengo et al. 2021). The drought scenario was not restricted to the Pantanal domain, and yet the biome was the one that experienced a record of fire count. Amazonia and Cerrado biomes, traditional sources of smoke towards Pantanal (especially Amazonia), did not experience a similar level of enhancement in fire activity, in 2020.

Figure 3 below (Figure 4 at page 19 of the revised version of the manuscript) shows that typically mean AOD over Pantanal strongly responds to the mean AOD over Amazonia, and it used to be similar or a fraction of mean AOD over Amazonia. In 2020 that was not the case, when the mean AOD value over Pantanal substantially surplused that over the Amazon, during both September and October. This aspect along with the enhancement of fire activity within the Pantanal domain were those that can be highlighted as significant or unprecedented when compared with the previous analyzed years and that can be assumed as anomalies. The revised version presents this discussion at page 6 lines 174 to 177.

■Is the smoke anomalous only in terms of aerosol loading, or have intrinsic optical properties changed?(Pag. 7 - Lines: 211 -221, Pag. 20/21 Figure 5 and 6)

R: It is difficult to answer this question focusing on Pantanal, since there was not an operational AERONET station or other measurements of aerosol intrinsic properties within Pantanal in 2020 (as far as we are aware). However, it is possible to do some evaluation

analyzing AERONET stations distributed across the regional smoke domain to get some insight on this matter, which we did, and is presented below. Focusing on the closest AERONET site to the Pantanal border, the Cuiaba site, we did not find a significant statistical difference between monthly mean SSA in 2020 and previous years (Figure 4 below and Figure 5, page 20 of the revised version). The same was observed for the other sites of AERONET analyzed. See discussion at page 7, lines 211 to 221 of the revised version of the manuscript.

■What effect does the anomalous smoke have on the radiative balance and what consequences does this have?

As discussed before, no statistically significant difference was observed in the data from the AERONET sites, either in the radiative balance. When analyzing the regional effect of the smoke layer on the radiative balance from CERES, 2007 presented the strongest attenuation of downward solar irradiance reaching the surface, much higher that 2020 regional smoke plume. The new version of the manuscript included this discussion at page 8, lines 250 to 256.

■How anomalous is this activity, not only in recent memory, but over scales spanning generations?

R: There was no need to go further into the past to show that, from a regional perspective and when compared with the mean scenario (within the period 2003-2020), 2020 biomass burning season could not be identified as an exceptional anomalous year, either based on fire count or smoke loading. As illustrated in Figure 1, the years 2004, 2007 and 2010 were more polluted and presented higher fire counts (Figure 2), regionally speaking (Amazonia+Cerrado+Pantanal).

However, when focusing specifically on the Pantanal domain, there are two informations that exceptionally differ from typical values (considering the period 2003-2020): the fire count observed in September and the mean AOD over the Pantanal biome during October (Figure 2). The high fire count over Pantanal in September 2020 has not been seen in previous years. As illustrated by the numbers cited before, in 2020, the fire count over Pantanal was 3.6 times higher than the mean climatological value. The manuscript indicates those changes at page 5, lines 141 to 161 and pages 7 and 8, lines 226 to 249.

■What caused the anomalous fire activity in the Pantanal?

R: Fire activity in Brazil biomes historically has a strong relationship with mankind intervention and there is a vast literature supporting this. Therefore, 2020 fire activity in Pantanal and the exceptional anomalous fire activity observed had, as usual, the mankind component. However, recent researches support that mankind traditional intervention was propelled by two distinct aspects:

a) A fire-prone environment (climate extreme, Marengo et al. 2021, Libonati et al., 2020). According to Marengo et al. (2021), the years of 2019 and 2020 were characterized by the worst drought in 50 years in Pantanal. The accumulated precipitation during the wet season of these years was between 50 and 60% less than normal.

b) An unfavorable governance (poor management and lax laws). According to Libonati et al. (2020), a combination of climate extremes, poor management and lax laws was behind Pantanal anomalous fire activity. Outdated environmental regulations, slashing of resources for environmental protection and climate actions in recent years certainly contributed to build-up the mentioned unfavorable governance.

One must also point out that the two concurrent aspects were not exclusive to Pantanal in 2020, and yet the fire count figures across Cerrado and Amazonia were not exceptionally far from those of recent years. Therefore, there are still open questions about the specific behavior of mankind intervention in Pantanal in 2020, a lack of studies of human causes and responses to fires in the Pantanal has been recognized as a challenge to a full comprehension of what happened (Libonati et al., 2020). See page 10, lines 300 to 310 of the revised version of the manuscript.

The authors present analysis that address most of these bullet points. There's a significant paper in this work, but I have to say that they don't pull the analysis together in a way that clearly provides the answers. Because of that I will recommend Major Revisions.

I have no need to remain anonymous. This is Lorraine Remer writing.
Thank you very much Dr. Remer, for your time and careful review! We really appreciated that!
Points to address are as follows:

- What is meant by **smoke anomaly** in this study? (a) Is it the overall smoke loading over the entire continent? (b) Is it just the smoke over the Pantanal? (c) Is it the smoke over the population centers of the Brazilian southeastern coast? At times, while reading, I had the feeling that the authors meant it to be (a), and then (b) and then (c). All of these are interesting, but the authors need to clarify when they are considering each one.

Let's assume that the main point is (b) because that is what Figure 6 addresses, although Figure 7 is more tuned to (a). Then I'm going to ask, "Why?". Why do we care specifically about how much smoke is over the Pantanal? It is a very small area from the regional perspective. I would think the question of "how much do Pantanal fires contribute to the regional big picture" to be the more interesting question rather than, "is the smoke above the Pantanal due to local or transported smoke". I mean, both questions are interesting, but the big picture is the bigger picture. If the authors find smoke and its consequences directly over the Pantanal to be the primary question to address, then they need to introduce the reason for this in the introduction… "The Pantanal represents a unique island of biodiversity in the region and smoke hanging over this area for up to six weeks has the potential for diminishing surface shortwave flux, stopping photosynthesis, interfering with primary productivity that has consequences as it cascades up the ecosystem." Or something like that. I know that that this is touched on here and there, but the paper needs to be structured in a way that makes this the primary focus.

If the authors are indeed looking more at the big regional picture (a) then there needs to be analysis presented " XX% of regional smoke is produced by the Patanal, representing only yy% of the regional surface area."

Or something like this. Or the authors could go in both directions. The paper is short. It could support two specific sections, one addressing (a) and one addressing (b). I don't need it to do both. I just need some clarity and focus communicated.

R: Although one could focus on one or another item, and indeed we are slightly tuned to Pantanal, the current version helps us to understand the need and importance of contextualizing both: (a) the 2020 overall smoke loading over the continent. (b) the 2020 smoke over the Pantanal. Thus, we show that regionally 2020 was not a particular year from the perspective of the plume dimension, loading and optical properties, and we explore what happened to Pantanal in 2020. In this sense, the focus on Pantanal presented more interesting results and analysis. And we adjusted the manuscript to highlight

the importance of Pantanal (locally and regionally) to support our focus and to show how protected areas (as indigenous and reserves) were atypically burned in the biome in 2020, as shown in Figure 3 (below and Figure 4 at page 19 of the new version of the manuscript) and further discussion ahead. In the revised version, the text was modified at pages 2 and 3, lines 56 to 70.

2.      Is there any insight gained from direct scatter plots of smoke vs. fire counts, and SW flux vs. AOD? Scatter plots of monthly means, for example, taking the points shown in Figure 2 and just throwing them into scatter plots. 4 month x 6 years. That's a 24 point scatter plot. The more fire counts, the more AOD, right? But if the Pantanal is more affected by advection than by local fires, there won't be much correlation. And maybe 2020 stands out, as an outlier. I don't know. It's just that right now the only thing I gain from Figure 2 is that 2020 is weird for both fire counts and smoke in the Pantanal, but that smoke weirdness lags fire weirdness by one month. There are a lot of words describing this figure, but few of those words point to the focus of the study.

R: The scatter plots of smoke vs fire counts over Pantanal and of smoke over Amazonia vs smoke over Pantanal indeed provided interesting insights (Figure 3 below and Figure 4 at page 19 of the new version of the manuscript). Smoke vs fire counts over Pantanal shows that, in general, the higher the fire count is, the higher the AOD, but there are several exceptions (ex. August 2020). The scatter plot smoke over Amazonia vs smoke over Pantanal  shows that the smoke over Pantanal has a stronger relationship with smoke over Amazon than with fire counts within the biome, suggesting that AOD over Pantanal is more affected by advection than by local fires. However, 2020 does stand out as an outlier. Typically, mean AOD over Pantanal domain is similar or a fraction of that over Amazonia domain. That was not the case for September and October of 2020, when mean AOD over Pantanal was much higher than over Amazonia, an indication that local smoke played an atypical role to the smoke level over Pantanal.

Regarding the one month lag between fire counts and AOD values, we added an analysis of fire count distribution on top of the vegetation index (EVI) within the Pantanal domain to help clarify a hypothesis that these observational data can point out (Figure 3 below and Figure 4 at page 19 of the new version of the manuscript). From plots (a), August was characterized by high fire count, but relatively low AOD, while October by relatively low fire count and high AOD. The high AOD over Pantanal in October, despite the reduction in fire count, hardly is explained by advection from Amazonia. As can be seen in the plot (b), there was a reduction in smoke loading over Amazonia during October, and the regional map of AOD for October (Figure 1 below and Figure 7 at page 16 of the new version of the manuscript) shows that there was a spot of high AOD centered and over the Pantanal domain. Therefore, a possible explanation for the lag between fire count and AOD from August to September/October could be the nature of the material being burned within Pantanal. As the maps show, during August a reduced number of fires was within conservation and Indigenous areas (where higher biomass density is present). However, from September on, there was a significant increase of fire number within these areas, which could explain the larger aerosol emissions and, consequently, the increase in AOD. The new version of the manuscript brings this information at pages 6 and 7, lines 195 to 210.

3.      Figure 3 is interesting because of the SSA, but the question I need answered is too hard to find in these plots. Is the SSA different in 2020 or not? Will radiative effects only be controlled by loading, or do changing optical properties play a role? Any thought of trying some 24-point scatter plots here also?

R: In order to answer the question, we replaced Figure 3 of the manuscript first version to a new one (Figure 5 of the current doc, Figure 6 at page 21 of the revised version), following the reviewer's suggestion. Instantaneous aerosol radiative forcing estimates from AERONET were plotted versus AOD at 550 nm, color coded by single scattering albedo (SSA) values, with data from July to October of the years 2003 to 2020. Data from 2020 are highlighted with red symbols. It is possible to observe that not only AOD but also SSA affect the downward solar irradiance at the surface and that no

difference can be noticed in 2020 data. To complement the information, boxplot of SSA at 440 nm for different sites and years are now presented in Figure 4 (Figure 5, page 20 of the revised version of the manuscript). From the boxplots, we conclude that SSA values from 2020 were similar to previous years. This information was added at page 7, lines 211 to 221 of the revised version.

4.        I thought Figure 4 was the most informative of the basic plots. Here you see the difference from year-to-year much better than in Figures 2 and 3. In 2020, the Pantanal stands out quite a bit darker than its immediate surroundings. This is the first place that I considered that local smoke might dominate AOD over the Pantanal. The authors also rightly point out the difference in flow between 2017 and 2020 that explains why the population centers of the southeastern coast were spared in 2017.

R: Despite the recognition of the informative value of this plot, following the new arrangement aiming to better contextualize the 2020 regional smoke plume and the smoke loading over Pantanal, the previous plot (Figure 4 of the first version submitted and mention in the reviewer question) was replaced by a similar analysis (Figure 1 of the current doc, Figure 7 at page 22 in the revised version of the manuscript) focusing on the intercomparison of 2020 regional smoke with the most polluted years (2004, 2005, 2007 and 2010) within the timeframe analyzed (2003-2020). The intercomparison now also includes maps from the month of October to highlight the high level of smoke restricted to the Pantanal domain and surroundings, which is evaluated as the most significant deviation from the historical perspective here analyzed (2003-2020), and in the context of Pantanal biome (Figure 2). In September 2020, despite the exceptional fire count level, the smoke loading over the Pantanal biome was not unprecedented. September of 2007 and 2010, for instance, presented AOD over Pantanal much higher than September 2020.

Related to the flow of the regional smoke plume towards the population centers of the southeastern coast, the new plot evidenced that this has also been seen in the past. For example, in 2004 and 2005 the monthly flow patterns were also towards the highly populated centers in the southeast of Brazil (Figure 1). See pages 7 and 8, lines 226 to 249 of the new version of the manuscript, where this discussion is presented.

5.        Figure 5. Have the authors considered an anomaly plot instead of absolute irradiance? Interannual differences are hard to see now.

R: Following the reviewer's suggestion, absolute irradiance was replaced by anomaly considering the period from 2003 to 2020 as reference to define the month climatology (Figure 8 of the modified version). However, as described and justified in the reply to reviewer query number 4, the focus has been changed to the intercomparison between 2020 and the most polluted regional smoke plume (2004, 2005, 2007 and 2010) in the decades. We expect that now it is much clearer to highlight the effect of 2020 regional smoke plume on surface solar radiation (SSR), how it stands when compared with the top polluted biomass burning season in the timeframe analyzed (2003-2020). September of 2020 anomalies in SSR are comparable to some of the most polluted years, but September 2007 presented the highest anomalies. Considering the atypical amount of smoke over Pantanal during october of 2020, we decided to include October in the comparison. The irradiance anomaly is discussed in Section 3.2, Page 8, Lines 250-256 of the newest version of the manuscript and the new results are shown in Figure 8.

Figure 6. Another good plot. It was very important to do the longer time series. Knowing that 2020 still had unprecedented fire counts in smoke, we see that the anomaly isn't that much greater than in the 2000s. People's memories are short. They forget how bad things were in the previous generation. Although, I didn't see anything interesting in the SW flux plot and I recommend dropping it. Also, shouldn't there be fire counts that go back another decade? It's not necessary, but it would be interesting. Fire counts were intense in the late 1980s and 1990s.

**R:** Pantanal 2020 BB season was indeed the one presenting relevant deviations from historical features, at least within the analyzed period in this manuscript (Figure 2) and according to INPE longest time series of fire count, which started in 1980's, as pointed out by Marengo et al. (2021). We also decided to include the same time series plots for Amazonia and Cerrado biomes. So one can have a better overview of the comparison between the biomes. The interannual and intraseasonal variability is now much more broadly discussed in the modified version of the manuscript, and the extended analyses previously described were included in Section 3.1, Page 5, Lines 141-161.

6.      Figure 7. I like this one also. This is a very nice summary of the entire situation over time.

**R:** Thank you, we kept this summary. In the newest version this is in Page 18, Figure 3.

7.      Figure 8. I couldn't understand this at all. I couldn't find in it the things mentioned in the text.  Maybe it is just me.

**R:** This plot and discussion aimed to explore the role of circulation interannual variability, via wind components. However, given that it did not present clear information, and considering that more new plots were added to the analysis, we understand that this plot may be excluded to make space for further discussion required by the new added plots. In the newest version of the manuscript the former Figure 8 was excluded.

8.      The three biomes, Amazonia, Cerrado and Patanal are mentioned very early in the introduction, before the map, sort of with the expectation that the reader already knows what they are. I suggest describing each one, as soon as they are mentioned. One thing that I struggled with is the relative sizes of these biomes. Patanal is so tiny. Why does anybody care?     Also,        Caatinga, Pampas, Mata Atlantica are mentioned with the expectation that the reader knows what these are, and we don't.

**R:** The description and location of the biomes are reorganized, including Caatinga, Pampas and Mata Atlantica, so the reader is better contextualized. A new map was developed (Figure 1 in the modified version of the manuscript) to clearly depict these biomes. In the newest version of the manuscript these modifications are in Section 2, Page 3, lines 72-77 and in Page 16, Figure 1. Indeed, Pantanal size is much smaller than the other biomes, however its unique biodiversity and its role to the regional hydrological cycle are of great relevance. The importance of the Pantanal biome is now emphasized in Section 2, Pag. 3, L81-85 of the modified version of the manuscript.

9.      Lines 85-90. We need more information on the surface irradiance. What is meant by cloud free in terms of footprint size? What is used for aerosol model to make the calculations? If there were to be a change in SSA, would the irradiances that are analyzed reflect that information? (Pag. 4 Lines 113-123)

**R:** Clouds and aerosol properties are obtained from MODIS with 10km nadir resolution and then projected (with the corresponding weights) into CERES SSF Level 2 footprint of 20 km (point-spread-functions are used to convolve higher resolution data into CERES 20km optical footprint). Finally, to obtain the SSF Level 3 product, the results are interpolated into 1º×1º latitude/longitude grid size. In this work a cloud-free scene is considered a scene that has aerosol retrievals on a given grid cell. According to CERES SSF 1 degree documentation, clear-sky parameters are only computed if cloud fraction is less than 0.1% in a given footprint.

(https://ceres.larc.nasa.gov/documents/DPC/DPC_current/pdfs/DPC_SSF1deg-Day_R5V1.pdf).

The aerosol model used in CERES surface irradiance calculations is very briefly described in Kratz, 2020. It uses near-real time daily AODs from the Model for Atmospheric Transport and Chemistry

(MATCH). The Optical Properties of Aerosols and Clouds (OPAC) Global Aerosol Data Set (GADS) was used to obtain the intrinsic aerosol properties, such as single scattering albedo and asymmetry parameter (Hess et al., 1998).

The complete information on cloud-free scenes in terms of footprint size and on the aerosol models used in CERES SSF products are in Section 2, Page 4, Lines 113-123 of the newest version of the manuscript.

10. Line 92. What is meant by Aqua bouncing?

R: We believe there was a mistranslation here. What actually happened was a problem with the data formatter for the solid state recorder aboard the Aqua satellite that resulted in a data acquisition gap from August 16 through September 2, 2020 for MODIS Aqua products.

Refs.: https://lpdaac.usgs.gov/news/aqua-satellite-anomaly/

https://landweb.modaps.eosdis.nasa.gov/cgi-bin/NRT/displayCase.cgi?esdt=MYD&caseNum=PM_MYD_20229_NRT&caseLocation=cases_data

Ultimately, in the newest version of the manuscript this information was not necessary, since we are now focusing only on the September and October maps of irradiance. Therefore, since the data gap that occurred mostly in the second half of August 2020, this technical problem did not significantly affect the new results.

11. Lines 244-247. These are some of the most important statements in the manuscript. Somehow the paper should be structured to get there. Also, this is where I started wondering about a scatter plot.

R: We added new plots and performed a reorganization in the discussions to accomplish this. See text at Results, pages 7 and 8, lines 195 to 210 and at Conclusions, page 9, lines 268 to 284.

12. Lines 252-253. Suddenly the authors start using the word "emissions". There are no emissions analyzed. The authors are working with fire counts, which are not the same as emissions. Then they state that the contribution of emissions from the Pantanal rivals that from Amazonia and Cerrado. In absolute terms Figure 2 shows that the total fire counts from the Pantanal never reaches the total numbers of the other biomes (because of its small area). Where else do the authors find evidence to support that statement?

R: The reviewer is right, the word emissions should not be used here, and there is no evidence to suggest that emission from Pantanal's rivals that of Amazonia and Cerrado in 2020. What can be suggested or inferred, based on fire counts, is that the relative contribution of Pantanal's emission to the regional smoke plume in 2020 was higher than the other analyzed years, given that the high amount of smoke over the biome in 2020 was associated with an explosion in local fire, especially during September and October. In the newest version of the manuscript the word "emissions" was removed and the text was modified to summarize the results that show the contribution of Pantanal to the regional smoke plume in 2020, in the Conclusion, Page 9, Lines 268-273.

13. I also note that a lot of statements found in Section 3.3 should really go into the Conclusions, especially Lines 244 and beyond.

R: The indicated statements were moved to the Conclusions, page 9, lines 268 to 284.

14. The Conclusions are mostly strong and well-stated. There's one sentence in the Conclusions

that was not supported in the body of the manuscript. "For Pantanal, 2020 was a very particular year, not exactly due to the aerosol loading over the biome, since in September of 2007 the biome experienced a higher monthly mean AOD, but due to the contribution of the local fire emission to the regional smoke plume. "

**R:** With the new analysis performed based on scatter plots of fire count vs smoke over Pantanal and AOD_Amazonia vs AOD_Pantanal (Figure 3 below and Figure 4 at page 19 of the revised version of the manuscript), we understood that now we are able to support this adjusted sentence:  *"For Pantanal, 2020 was a very particular year, not exactly due to the aerosol loading over the biome, since in  September of 2007 the biome experienced a higher monthly mean AOD, but due to the contribution of local fire emission to the local smoke plume, which surpassed advection from Amazonia biomass burning areas".* See discussion at page 9, lines 273 to 283 of the revised version.

There is no analysis that states that the Pantanal contributes XX% of the smoke in the regional smoke plume. Even after all of this analysis, this statement is still based on qualitative discussion. What has this study contributed to supporting this statement that the NYT, BBC and Le Monde have not?

R: Yes, that's right, current analysis is not able to quantify Pantanal contributions to the regional smoke plume. However, with new observational plots and analysis characterizing Pantanal contribution to the level of smoke compared with the advection from Amazonia and the intraseasonal variability in the nature of areas being burned and its impact on the amount of smoke (See discussion at pages 6 and 7,Line 195 -210, Figure 4 at pag. 19) , we hope that the manuscript is providing a better and more consistent discussion about Pantanal 2020 bb season. Additionally, we provide a contextualization of 2020 bb regarding previous years for both Pantanal and regional perspectives. (See discussion at page 5 Lines: 141-161 and Figure 2, Pag 17)

15.     Finally, and the most frustrating… Why did the Pantanal burn in 2020? What is fundamentally different about this year that fire erupted in a way unseen for over a decade? Drought? Human intervention or non-intervention? Why? While the paper can be published without answering this question, it is the question in the reader's mind as they read through the analysis. Why the fire eruption in the Pantanal in 2020? There's an elephant in the room that nobody is mentioning. A paragraph in the Conclusions with some educated hypotheses in a Discussion format would be satisfying.

 R: As discussed above, the most reliable hypothesis about 2020's Pantanal fires is that it was due to a combination of a drought-prone environment and lack of effective public policies to protect the environment against man-made fires. Actually, unfortunately there was a huge throwback on the steps previously taken regarding environmental laws, additionally resources for environmental protection and climate actions have been slashed. During the last years, particularly after the 2018 elections, Brazilian authorities have adopted against-the-environment and pro-agricultural-livestock-expansion speeches. Under the new government, Brazil's main environmental enforcement agency, Ibama, is weakened and it reduced the number of fines for those involved with deforestation. In this context, indigenous and conservation areas have become exposed targets for those aiming to expand illegally agricultural land. As suggested, a paragraph was included in the Conclusions, Page 10, Lines 300-313 in the modified version of the manuscript to address this important issue.

(a)

[Figure]

(b)

[Figure]

Figure 1: Regional Smoke Plume (a) September ; (b)October

[Figure]

Figure 2: Fire count and AOD@550 nm interannual variability (a) Pantanal; (b) Amazonia; (c) Cerrado

[Figure]

Figure 3: (a) AOD@550 nm Pantanal versus Fire count Pantanal; (b) AOD@550 nm Pantanal versus AOD@550 nm Amazonia; (c) Fire spots distribution across Pantanal for August, September and October of 2020 using Enhanced Vegetation Index(EVI) as background and highlighting Indigineous and conservation areas; (d) Pantanal total fire count and fire count within Conservation and Indigenous areas.

[Figure]

Figure 4: Interannual variability of mean Single Scattering Albedo (SSA) during biomass burning season (Aug-Sept-Oct) from AERONET stations (Please, refer to Figure 7 to see the site geographical distribution).

[Figure]

Figure 5 - Aerosol direct instantaneous Radiative forcing as function of AOD at 550 nm at AERONET sites highlighting 2020 (red) against historical value (2003 – 2019)

**(a)**

[Figure]

Figure 6 - Surface Solar Radiance anomaly: (a) September; b) October

[Figure]

Figure 7 - New map representing Brazil's biomes and AERONET sites.

**References:**

LIBONATI R ET AL. 2020. Rescue Brazil's burning Pantanal wetlands. Nature 588: 217-219. https://doi.org/10.1038/d41586-020-03464-1.

Marengo, J. A. et al. Extreme drought in the Brazilian Pantanal in 2019–2020: Characterization, causes and impacts. Front. Water3, 639204 (2021).

**RESPONSE TO REVIEWER 02**

We would like to thank the reviewer for his/her important and meaningful questions. The replies to the questions, comments and suggestions are in blue color and the provided figures and references are displayed at the end of the current document. The figures presented here have been used to improve the manuscript's new version. The place and references of the changes in the new manuscript version to respond to the reviewer are marked in yellow.

Comment on acp-2021-1086:

The paper has identified the irregularity of the biomass burning in the Amazon Forest for 2020 compared to the previous 6 years, which has involved the Pantanal biome. Evidence of the hotspot counts, AOD from MODIS and AERONET and retrieved solar irradiance have clearly shown the burning anomaly of the year. The statements are well discussed and proved but the contributing factors of the anomaly are not discussed in the paper.

Firstly, there is less specification on the main anomaly that is discussed whether it is a burning or emission or something else? Secondly, the factors that cause these anomalies should be understood, such as reason of burning (natural forest fire/human activities), source of burning (where and type of land cover burnt), weather condition (local or transported emission/conducive to sustain burning/conducive to fire spread). Since the year of 2020 is the COVID year, please do mention whether it played a role in the 2020 burning condition. A good background information would be helpful for the audience and further analysis.

R: First, we would like to clarify that, following both reviewers comments, we reassess the identification of anomalies in the context of Pantanal and Brazil 2020 biomass burning season. We change the time frame of the analysis, looking also for the other biomes (Amazonia and Cerrado, not just Pantanal), for the period between 2003-2020.

Looking further into the past, it shows that, from a regional perspective and when compared with the mean scenario (within the period 2003-2020), 2020 biomass burning season could not be identified as an unprecedented and exceptional anomalous year, either based on fire count or smoke loading. As illustrated in Figure 1 (below, which was incorporated as Figure 7 at page 16 of the new version of the manuscript), the years 2004, 2007 and 2010 were more polluted and presented higher fire counts (Figure 2 below and at page 17 of the revised version of the manuscript), regionally speaking (Amazonia+Cerrado+Pantanal). Therefore, regionally, the focus has been changed to the intercomparison between 2020 and the most polluted regional smoke plume (2004, 2005, 2007 and 2010) occurred between 2003 and 2020. The extended discussion was included at page 5, lines 141 to 161.

However, when focusing specifically on the Pantanal domain, there are two aspects evidencing that 2020 exceptionally differs from typical years (considering the period 2003-2020): the fire count observed in September and the mean AOD over the Pantanal biome during October (Figure 2). The high fire count over Pantanal in September 2020 has not been seen in previous years. Since there is already a detailed study on the climate conditions and its potential role on the Pantanal 2020 Biomass burning season (Marengo et al. 2021) we try to better understand the relationship between smoke levels over Pantanal and local fire count and smoke in the Amazon, from where transport to Pantanal

is important (Figure 3 below, Figure 4 at page 19 of the revised version of the manuscript).

To contextualize the factors associated with Pantanal exceptional fire count in 2020, as background information for the new version of the manuscript, we cited Marengo et al. (2021) and Libonati et al. (2020) studies as follow (at page 6, lines 174 to 177 and page 7, lines 195 to 221 of the revised version of the manuscript):

Fire activity in Brazil biomes historically has a strong relationship with mankind intervention and there is a vast literature supporting this. Therefore, 2020 fire activity in Pantanal and the exceptional anomalous fire activity observed had, as usual, the mankind component. However, recent researches support that mankind traditional intervention was propelled by two distinct aspects:

  a) A fire-prone environment (Marengo et al. 2021, Libonati et al., 2020). According to Marengo et al. (2021), the years of 2019 and 2020 were characterized by the worst drought in 50 years in Pantanal. The accumulated precipitation during the wet season of these years was between 50 and 60% less than normal.

  b) An unfavorable governance (poor management and lax laws). According to Libonati et al. (2020), a combination of climate extremes, poor management and lax laws was behind Pantanal anomalous fire activity. Outdated environmental regulations, slashing of resources for environmental protection and climate actions in recent years certainly contributed to build-up the mentioned unfavorable governance.

One must also point out that the two concurrent aspects were not exclusive to Pantanal in 2020, and yet the fire count figures across Cerrado and Amazonia were not exceptionally far from those of recent years. Therefore, there are still open questions about the specific behavior of mankind intervention in Pantanal in 2020, a lack of studies of human causes and responses to fires in the Pantanal has been recognized as a challenge to a full comprehension of what happened (Libonati et al., 2020).

In a recent study (Vale et al., 2021), the conclusion was that the current administration took advantage of the COVID-19 pandemic to intensify a pattern of weakening environmental protection in Brazil. The study examined the effects of the pandemic on environmental protection and legislation in Brazil in the current administration and showed 57 legislative acts aimed at weakening environmental protection, almost half of which in the seven-month period of the pandemic in Brazil. The study also found a 72% reduction in environmental fines during the pandemic, despite the increase observed in Amazonian deforestation during the analyzed period. It is important to stress that this context encouraged people to set fire around the country, particularly in indigenous and protected areas. This background is being considered in the new version of the manuscript. (See page 10, lines 300 to 313 of the revised version of the manuscript).

In the result analysis (Section 3.2, 3.3), the analysis time frame of the data should be consistent. Hotspot, AOD and solar radiance data from 2003-2020 (Figure 6 – 8) were used for the chronological change of the burning events in Pantanal biome, but not for the spatial map (Figure 4 – 5) where the anomaly is identified. The inclusion of long-term dataset could assist on the understanding of spatial distribution of the burning condition more clearly.

R: In the revised version of the manuscript, we kept the time series plots of fire count and AOD (Figure

2 below and at page 17 of the revised version of the manuscript) from 2003 to 2020, and we extended it to Cerrado and Amazonia biome, but the time frame was changed for the spatial maps. We included maps for September and October of the more recent years (between 2003 to 2020) identified as the most polluted in terms of the regional smoke plume (Figure 1) to be compared to the results for 2020 (2004, 2005, 2007, 2010, 2017), the modified text can be found at page 5, lines 141 to 161. So, instead of plotting all the maps of September from 2003 to 2020, we focus on the most polluted years and add maps for October to highlight the atypical high level of smoke over Pantanal at the end of the bb season. We did it for both, AOD (Figure 1) and Surface Solar Radiance anomaly (Figure 4 below and Figure 8 at page 22 of the revised version).

In the discussion section (Section 3.3) 2020 has been referred to the previous burning season in 2000s several times, in terms of the similarity of hotspot amount and AOD level. No detailed information is provided on the cause of the burning, even there are multiple similar occasions that have had happened in the past, which might not be so much of an anomaly but reoccurrence if longer period data (2003-2020) is considered.

**R:** The reviewer is corrected and the revised version of the manuscript tries to make it clearer what the authors considered anomalous in 2020. In fact, in terms of the regional smoke plume, in 2007 higher AOD values were detected and covered a larger area in South America. As mentioned previously, in Pantanal, however, in 2020, the fire count detected in September and the mean AOD over the Pantanal biome during October differ from previous years values. So, regionally (Brazil), within the time frame analyzed, 2020 was not different from past bb season (Figure 1) in terms of smoke loading and total fire count, it was Pantanal that presented aspects not seen in the previous years, as mentioned. The modified text can be found at page 5, lines 141 to 161.

Overall, the paper is not well structured, and information are clumped together in long paragraphs. Please split the lengthy paragraph or introduce sub-section for clarity. A major revision is required before the paper is deemed suitable for acceptance and publication.

**R:** We reformulate the structure of the paper, we expect that the revised version is better structured, making the readability easier. See pages 2 and 3, lines 56 to 70.

Specific comments:

Line 56-60: Please provide the basic information on the size and the typical cause of burning in the different biomes. It would be helpful to elaborate the uniqueness of burning/emission condition in Pantanal.

**R:** In general, slash and burn practices are tools in the deforestation process of the original vegetation. Fires are also set to clear the land from residues from the previous crop season. Particularly in 2020, analyzing fire count map over EVI map, mankind intervention over indigenous and protected areas was clearly identified (Figure 3 below, Figure 4 at page 19 of the revised version of the manuscript). The social and economic motivations, specially during 2020 season, need to be better studied but they are out of the scope of the present work, although as pointed before, the current administration's efforts to weaken environmental protection in Brazil (Vale et al., 2021), lax laws and lack of fiscalization must have contributed to this scenario (Libonati et al, 2020). See page 10, lines 300 to 313 of the revised version.

Line 222-223: How about 2003?

R:  Yes, 2003 could be included since it is the beginning of the time series, nevertheless, the fire count and AOD over Pantanal in 2003 were much lower than those identified as polluted. This has been adjusted in the new version of the manuscript, see the new version of Figure 2 at page 17 and discussion at page 7, lines 195 to 221.

Line 234-237: The role of weather anomaly is mentioned here but just for year of 2015-2016. How about the other year of extreme burning between 2003-2010 (Line 222-225)? More explanations need to be provided.

R: Considering the new arrangement, to focus the spatial comparison between 2020 and the most polluted years between 2003 and 2020, what we found in common among the polluted years (including 2020) is the prevalence of drought conditions over the center-west region of Brazil and southern Amazonia (Figure 5).  As in 2020, the drought conditions also spread across Pantanal in 2010. This again points out  to the particularity of the human component  in the explosion of fire in September of 2020. This has been addressed in the new version of the manuscript, at page 6, lines 174 to 177 and page 10, lines 300 to 313.

(a)

[Figure]

(b)

Figure 1: Regional Smoke Plume (a) September ; (b)October

[Figure]

Figure 2: Fire count and AOD@550 nm interannual variability (a) Pantanal; (b) Amazonia; (c) Cerrado

[Figure]

Figure 3: (a) AOD@550 nm Pantanal versus Fire count Pantanal; (b) AOD@550 nm Pantanal versus AOD@550 nm Amazonia; (c) Fire spots distribution across Pantanal for August, September and October of 2020 using Enhanced Vegetation Index(EVI) as background and highlighting Indigineous and conservation areas; (d) Pantanal total fire count and fire count within Conservation and Indigenous areas.

**(a)**

[Figure]

Figure 4 - Surface Solar Radiance anomaly: (a) September; b) October

[Figure]

Figure 5 - Brazil Standard Precipitation Index (SPI) in September for the 3 month scale and for the years 2004, 2005, 2007, 2010, 2017 and 2020. Red areas represent extremely dry conditions (source: http://clima1.cptec.inpe.br/spi/pt)

**References:**

LIBONATI R ET AL. 2020. Rescue Brazil's burning Pantanal wetlands. Nature 588: 217-219. https://doi.org/10.1038/d41586-020-03464-1.

Marengo, J. A. et al. Extreme drought in the Brazilian Pantanal in 2019–2020: Characterization, causes and impacts. Front. Water3, 639204 (2021).

Vale, M. M., Berenguer, E., Argollo de Menezes, M., Viveiros de Castro, E. B., Pugliese de Siqueira, L., & Portela, R. (2021). The COVID-19 pandemic as an opportunity to weaken environmental protection in Brazil. Biological conservation, 255, 108994. https://doi.org/10.1016/j.biocon.2021.108994

---

## Author Response (AR3)

**Suggestions for revision or reasons for rejection (will be published if the paper is accepted for final publication)**

Reviewer comments - in black
*Authors Reply (R:)- in blue*

The paper now address a concise scientific question with very good analysis and excellent plots. Scientifically it is ready for publication. I have no issues with the study, the plots, the conclusions or the structure of the paper. It is really excellent.

But, the writing itself is weak. I started to compile a list of issues, but this only scratches the surface. I understand what the authors are saying, but many times I have to read the sentence twice to get the information. Because of the writing, I would not categorize the paper as acceptable 'as is'.

**We would like to express our gratitude to Dr. Remer for her careful reading and relevant contributions to the improvement of this manuscript. We recognized their relevance and accepted all the recommendations and suggestions.**

**Q1. Lines 14-16**. Aiming to contextualize the regional and Pantanal burning seasons, the present study analyzes fire counts and smoke over Pantanal, Amazonia and Cerrado in 2020 with the previous seventeen years (2003-2019). Sentence doesn't make sense.

**R: The sentence was adjusted (as can be seen below) to express in clearer way the goal of the manuscript** ==(See manuscript new clean version Lines 14-15)==

*"The present study analyzes fire counts and smoke over Pantanal in 2020, comparing this particular year's data with those from the previous seventeen years (2003-2019)"*

**Q2. Lines 24-25** The nature of the burned areas in Pantanal was determinant to the intraseasonal variability of the smoke within the biome in 2020.

What does determinant mean here.

**R: The sentence was adjusted, as follow, to better express its meaning** ==(See manuscript new clean version Lines 23-25).==

*"The observed intraseasonal variability of smoke over Pantanal revealed to be largely driven by the nature of the burned areas in the biome."*

**Q3. Line 37**. Its perturbation on the regional climate span from lowering the solar energy availability

**R: The sentence was adjusted to improve the summary of the ways through which the Regional Smoke Plume affects the South America regional climate. (See manuscript new clean version Lines 38-40)**

*"The RSP affects the regional climate reducing the availability of solar energy at the surface and perturbing the cloud microphysics and atmospheric thermodynamics and chemistry."*

**Q4. Line 73**. South not southern Line
**R:** southern ***was replaced by*** south **See manuscript new clean version Line 73)**

**Q5. Line 77**. Exposition
**R:** *"Exposition"* **word was excluded and the text adjusted including the expression** *"it is less affected by"* **(See manuscript new clean version Line 77)**

**Q6. Line 84** critics
**R:** *"critics"* ***was excluded and the text was adjusted as follows*** *"Large scale biomass burning within its domain and the RSP affect the local and regional climate equilibrium and also pose a direct threat to the health of Pantanal's singular biodiversity and population."*
**(See manuscript new clean version Lines 83-85)**

**Q7. Line 85**. Pantanal to Pantanal's
**R:** *"Pantanal"* **was replaced by** *"Pantanal's"* **(See manuscript new clean version Line 85)**

**Q8. Lines 85-86**. For land use consideration of fire distribution across Pantanal during the 2020 burning season, highlighting conservation and indigenous areas, were used

**R: The sentence was adjusted (as can be seen below) to clarify its message. (See manuscript new clean version Lines 85-87)**

*"MODIS (Moderate Resolution Imaging Spectroradiometer) Enhanced Vegetation Index (EVI, Kidan 2021) was applied in land use characterization across Pantanal during the 2020 burning season, specially to highlight conservation and indigenous areas, which tend to present higher biomass density."*

**Q9. Line 106**. Extension or extent
**R: The term** *"distribution"* **was used instead of extension** *or extent, which is a much clearer term.* **(See manuscript new clean version Line 107)**

*"...to analyze the distribution of illegal biomass burning…"*

**Q10. Line 107-108**. The fire counts used, not the used fire counts

*R: "...the used fire counts…"* ***replaced by*** *"The fire counts used…"* **(See manuscript new clean version Lines 107-108)**

**Q11. Line 126**. South, not southern

*R: "southern"* ***was replaced by*** *"south"* **(See manuscript new clean version Line 126)**

**Q12. Line 132**. the regional plume influence dominium How about instead simply, the regional plume.

*R: "...the regional plume influence dominium."* ***replaced by*** *"the regional plume."* **(See manuscript new clean version Line 132)**

**Q13. Line 192-194** However, two interesting aspects worth emphasizing are that Pantanal, for October 2020, presented a much lower fire count compared to August and September, and higher smoke loading than Amazonia.

Confusing. Is the comparison between October and the previous months OR between Pantanal and Amazonia?

**R: The sentence was reviewed, as follows, and we hope that the explanation is much clearer now. (See manuscript new clean version Lines 190-194)**

*"Regarding the intraseasonal variability of smoke aerosol loading, October 2020 in Pantanal can be evaluated as exceptional when compared with October from the previous years of the time series here analyzed. An interesting aspect worth emphasizing for October 2020 is that Pantanal presented much lower fire counts compared to August and September (Figure 3). However, August's higher fire count in Pantanal did not translate into a high level of smoke over the wetland biome.*

**Q14. Line 261**, "about that" about what? What are the authors referring to?

**R: The sentence was modified (see below) in order to clarify what we would like to express. (See manuscript new clean version Lines 257-258)**

*"However, with the explosion of fire counts across Pantanal in 2020, there was a question about the role of Pantanal's smoke emissions to the RSP during the 2020 biomass burning season."*

**Q15. Line 277** revealed to be determinant to the amount It sounds less stilted to say "revealed to determine the amount"

*R: Suggestion accepted* **(See manuscript new clean version Lines 274-275)**

**Q16. Line 278** Fire counts instead of fire number

*R: Suggestion accepted* (See manuscript new clean version Line 276)

**Q17. Lines 297-298**. when one considers the climate projections of increasing the frequency of drought conditions and the role of an adequate governance

The use of the word governance is not common in this context and in prior uses. What are the authors trying to say? On one hand we need to consider the likelihood of increased drought, but what are we considering about governance? Are the authors trying to say that BECAUSE of the likelihood of more drought, policy makers will need to take measures to mitigate future scenarios similar to the one shown here in 2020. "Adequate governance" is ambiguous.

**R: We accepted the reviewer suggestion and adjusted the sentence in the manuscript as follows. (See manuscript new clean version Lines 294-295)**

*"Because of the likelihood of an increase in drought frequency and intensity, policy makers will need to take measures to mitigate future scenarios similar to the one shown here in 2020."*

**Q18. Lines 301-303**. Pantanal 2020 biomass burning season could be seen as an example of the worst scenarios combination, climate extreme related to a fire-prone environment (Marengo et al. 2021, Libonati et al., 2020) and an unfavorable governance (Vale et al., 2021).

This sentence has issues. THE Pantanal 2020 biomass burning season represents the worst combination of a climate extreme applied to a fire-prone environment, coupled with inadequately enforced environmental regulations.

It's that "governance" again. The word just doesn't make sense to a native American English speaker.

**R: The sentence was adjusted including the reviewer suggestion, and we expect that it is much clear now. (See manuscript new clean version Lines 298-301)**

*"...the Pantanal 2020 biomass burning season represents the worst combination of a climate extreme applied to a fire-prone environment, coupled with inadequately enforced environmental regulations (Marengo et al. 2021, Libonati et al., 2020, Vale et al., 2021)."*

**Q19. Lines 304 – 307**. Valet et al. (2021) pointed out a large reduction in environmental fines during the pandemic was identified, despite the observed increase in Amazonian deforestation, slashing of resources for environmental protection and climate actions in 306 recent years and approval of legislative acts aimed at deregulating and weakening environmental protection during the COVID- 19 pandemic.

Another sentence with an inability to properly express the points.

Valet et al., (2021) pointed out a large reduction in environmental fines during the pandemic. This was a decrease in fine enforcement, not a decrease in environmental violations, as there was an observed increase in Amazonian deforestation. The slashing of funding for environmental protection and climate actions during the pandemic years has compounded the environmental harm imposed by legislative acts aimed at degrading environmental protection.

**R: The sentence was adjusted (see below) considering the reviewer comments, and we expect that it is much clearer now. (See manuscript new clean version Lines 302-305)**

*"Vale et al. (2021) pointed out a large reduction in environmental fines during the pandemic. This was caused by a decrease in fine enforcement, not due to a decrease in environmental violations, as an increase in Amazonian deforestation was observed. The slashing of funding for environmental protection and climate actions during the pandemic years has compounded the environmental harm imposed by legislative acts aimed at degrading environmental protection."*